



# Molecular Composition of Particulate Matter Emissions from Dung and Brushwood Burning Household Cookstoves in Haryana, India

Lauren T. Fleming[1], Peng Lin[2], Alexander Laskin[2], Julia Laskin[2], Robert Weltman[3], Rufus D. Edwards[3], Narendra K. Arora[4], Ankit Yadav[4], Simone Meinardi[1], Donald R. Blake[1], Ajay Pillarisetti[5],
Kirk R. Smith[5], Sergey A. Nizkorodov[1]

[1]Department of Chemistry and [3]Department of Epidemiology, University of California, Irvine, CA 92617
[2]Department of Chemistry, Purdue University, West Lafayette, IN 47907
[4]The Inclen Trust, Okhla Industrial Area, Phase-I, New Delhi-110020, India
[5]School of Public Health, University of California, Berkeley, CA 94720

*Correspondence to*: Rufus D. Edwards (edwardsr@uci.edu) and Sergey A. Nizkorodov (nizkorod@uci.edu)

**Abstract.** Emissions of airborne particles from biomass-burning are a significant source of black carbon (BC) and brown carbon (BrC) in rural areas of developing countries where biomass is the predominant energy source for cooking and heating. This study explores the molecular composition of organic particles from household cooking emissions, with a focus on identifying fuel-specific compounds and BrC chromophores. Traditional meals were prepared by a local cook with dung
and brushwood-fueled cookstoves in a village of Palwal district, Haryana, India. The cooking events were carried out in a village kitchen while controlling for variables including stove type, fuel moisture content, and meal. The particulate matter ($PM_{2.5}$) emissions were collected on filters, and then analyzed via nanospray desorption electrospray ionization/high resolution mass spectrometry (nano-DESI-HRMS) and high performance liquid chromatography/photodiode array/high resolution mass spectrometry (HPLC-PDA-HRMS) techniques. The nano-DESI-HRMS analysis provided an inventory of
compounds present in the particle phase. Although several compounds observed in this study have been previously characterized using gas chromatography methods, a majority of species in nano-DESI spectra were newly observed biomass-burning compounds. Both the stove (*chulha* or *angithi*) and the fuel (brushwood or dung) affected the composition of organic particles. The geometric mean $PM_{2.5}$ emissions factor and the molecular complexity of $PM_{2.5}$ emissions increased in the following order: brushwood/*chulha* (4.9±1.7 g kg$^{-1}$ dry fuel, 93 compounds), dung/*chulha* (12.3±2.5 g kg$^{-1}$ dry fuel, 212
compounds), and dung/*angithi* (16.7±6.7 g kg$^{-1}$ dry fuel, 262 compounds). The lower limit for the mass absorption coefficient (MAC) at 365 nm and 405 nm for brushwood $PM_{2.5}$ was 3.4 m$^2$ g$^{-1}$ and 1.8 m$^2$ g$^{-1}$, respectively, which was approximately a factor of two higher than that for dung $PM_{2.5}$. The HPLC-PDA-HRMS analysis showed that, regardless of fuel type, the main chromophores were $C_xH_yO_z$ lignin fragments. The main chromophores accounting for the higher MAC values of brushwood $PM_{2.5}$ were $C_8H_{10}O_3$ (tentatively assigned syringol), possible nitrophenol species $C_8H_9NO_4$, and
$C_{10}H_{10}O_3$ (tentatively assigned methoxycinnamic acid).



# 1 Introduction

Approximately 3 billion people live in residences where solid fuels (coal, wood, charcoal, dung, and crop residues) are combusted for cooking (Smith et al., 2014). Approximately 57% of Indian households report use of wood (49%) or crop residues (9%) as their primary cookfuels, while 8% report dung as a primary cookfuel (Census of India, 2011). However, many households will routinely use two or more of these fuels for their cooking needs, often in combination, in simple, home-made traditional stoves, or *chulhas*. These biomass-burning cookstoves have low combustion efficiencies and produce significant emissions of pollutants, including fine particulate matter ($PM_{2.5}$).

The epidemiological literature statistically links household air pollution from solid biomass to acute lower respiratory infections in children; heart disease, stroke, cataracts, and cancers in adults, as well as low birth weight for infants of pregnant women (Smith et al., 2014). $PM_{2.5}$ are small enough to infiltrate deep into the lungs and penetrate the body's defenses, and therefore $PM_{2.5}$ exposure has been commonly used for estimating risks from both ambient air pollution and cigarette smoke (Finlayson-Pitts and Pitts, 2000). The degree of adverse health effects of cookstove smoke likely depends on the chemical composition of the $PM_{2.5}$, however, which is largely unknown (Araujo et al., 2008).

Household cooking is estimated to be responsible for 26-50% of ambient $PM_{2.5}$ in India (Chafe et al., 2014; Guttikunda et al., 2016; Lelieveld et al., 2015). Of this emissions mixture, carbonaceous particles affect climate directly by scattering and absorbing incoming solar radiation and indirectly by acting as cloud condensation nuclei (Crutzen and Andreae, 1990). In addition to black carbon (BC), which absorbs solar radiation across the entire visible spectrum, some molecules in biomass burning aerosol are brown carbon (BrC) chromophores, which efficiently absorb blue and near-UV solar radiation (Laskin et al., 2015). Modeling studies have shown that in certain geographic areas climate warming by BrC has the potential to outweigh cooling by scattering organic aerosols (Feng et al., 2013). South Asia has been identified as one of these unique regions where emissions from cookstoves are a significant source of regional BrC (Feng et al., 2013).

Cookstove emissions have been studied in both the laboratory and field settings. Field studies typically involve observations and measurements during daily cooking activities in rural village homes. For example, Xiao et al. (2015) measured BC and $PM_{2.5}$ throughout the day for 6 different houses to monitor indoor concentrations in the household (Xiao et al., 2015). Stockwell et al. (2016) utilized a photoacoustic spectrometer to conduct in situ absorption measurements at 405 and 895 nm to monitor BC and BrC emissions from cook fires in Nepal (Stockwell et al., 2016). With a literature-recommended mass absorption coefficient of $0.98 \pm 0.45$ m$^2$g$^{-1}$ at 404 nm (Lack and Langridge, 2013) and measured particle absorption by the photoacoustic spectrometer, they approximated emission factors (EFs) for BrC. BrC EFs were more than a factor of 1.5 higher for the hardwood smoke (EF=10.6 g kg$^{-1}$ fuel) compared to the dung smoke (EF=5.85 g kg$^{-1}$ fuel). Pandey et al. (2016) collected $PM_{2.5}$ on filters from cookfires in India, fueled by wood, agricultural residues, dung, and a mixture thereof and reported mass absorption coefficient (MAC) values (Pandey et al., 2016). They found that the MAC at 550 nm was a factor of 2.6 higher for fuel wood (1.3 m$^2$ g$^{-1}$) compared to dung (0.5 m$^2$ g$^{-1}$) (Pandey et al., 2016).





In the laboratory, water boiling test (WBT) protocols are utilized to evaluate stove performance (Global Alliance for Clean Cookstoves, 2014). The WBT standard protocols are made up of three phases to represent the stove's combustion efficiency while cooking: (1) high power, cold start (2) high power, hot start (3) low power, simmer (Global Alliance for Clean Cookstoves, 2014). While the WBTs can be carried out under more controlled conditions, recent studies have found that the

WBTs fail to capture periods of low combustion efficiency in cooking events (Chen et al., 2012; Johnson et al., 2008, 2009). This is due to daily cooking activities involving more than just boiling water (Johnson et al., 2009). Some cooking techniques require a smoldering fire, for example, the cooking of *chapatti,* a traditional Indian bread (Johnson et al., 2009). Alternately, these low combustion efficiency periods may be a consequence of multitasking around the home (Johnson et al., 2009). The literature estimates that emissions of $PM_{2.5}$ (Roden et al., 2009) and $CO/CO_2$ ratios (Johnson et al., 2008; Kituyi

et al., 2001; Ludwig et al., 2003) are underrepresented by the WBTs relative to field measurements by a factor of 3. There are also concerns that WBTs cannot be scaled to real cooking events and that climate models may underrepresent global emissions from biomass-burning cookstoves (Chen et al., 2012; Johnson et al., 2008, 2009).

The organic components of biomass burning organic aerosols (BBOA) have been successfully characterized in previous studies by electrospray ionization high resolution mass spectrometry (ESI-HRMS) (Budisulistiorini et al., 2017; Laskin et

al., 2009; Lin et al., 2012, 2016, 2017; Smith et al., 2009; Wang et al., 2017; Willoughby et al., 2016). For example, ESI-HRMS was used to analyze the particle-phase constituents of smoke samples collected during the Fire Lab at Missoula Experiment (FLAME) campaign (Laskin et al., 2009; Smith et al., 2009). Fuels utilized in the FLAME studies were selected to represent North American wild fires, while those presented in these publications were largely from non-woody biomass fuels such as detritus and litter as well as Southern Californian *ceanothus*. Smith et al., (2009) reported an inventory of

species in particle-phase BBOA, with 70 percent of compounds being reported for the first time. Laskin et al., (2009) examined the nitrogen-containing species, and observed that a large fraction of these species are N-heterocyclic compounds. Lin et al., (2016) identifies fuel-specific BrC chromophores in particles collected from the FLAME-4 experiments via high performance liquid chromatography/photodiode array/high resolution mass spectrometry (HPLC-PDA-HRMS). Two of the four fuels were woody biomass specific to North America. They found that nitroaromatics, PAHs, and polyphenols were

responsible for the light absorption by BBOA (Lin et al., 2016). The most recent papers investigated the chromophores in BBOA from Lag Ba'Omer, a nationwide bonfire festival in Israel (Bluvshtein et al., 2017; Lin et al., 2017). They found nitroaromatics to be the most prominent chromophores in these samples (Bluvshtein et al., 2017; Lin et al., 2017). Budisulistiorini et al. (2017) similarly identified 41 chromophores from Indonesian peat, charcoal, and fern/leaf burning with a method relying on chromatographic separation and simultaneous detection by spectrophotometry and ESI-MS

(Budisulistiorini et al., 2017). They identified three types of chromophores: oxygenated, nitroaromatics, or sulfur-containing (Budisulistiorini et al., 2017).

The goal of the current study is to understand the composition of cookstove BBOA in more detail than afforded by previous measurements. We do this by 1) generating and collecting BBOA from prescribing cooking events carried out by a local



cook, and 2) using high-resolution mass spectrometry techniques to characterize their particle-phase composition. It is part of a larger study attempting to document the contribution of household combustion to ambient air pollution in India.

In this paper we provide an inventory of particle-phase compounds detected by nano-DESI-HRMS, and an assessment of BrC chromophores specific to the biomass type used based on HPLC-PDA-HRMS analysis. For the first time, the chemical

composition of brushwood smoke is probed in detail and compared to the much less-studied dung smoke.

## 2 Experimental Methods

### 2.1 Field Site

This study was conducted at the SOMAARTH Demographic, Development, and Environmental Surveillance Site (Balakrishnan et al., 2015; Mukhopadhyay et al., 2012; Pillarisetti et al., 2014) run by the International Clinical

Epidemiological Network (INCLEN) in Palwal District, located approximately 80 km south of New Delhi. SOMAARTH covers 51 villages across three administrative blocks, with an approximate population of 200,000. Palwal District has a population of approximately 1 million over ~1400 km$^2$; 39% of residents in the district use wood as their primary cookfuel, followed by dung (25%) and crop residues (7%) (Census of India, 2011).

### 2.2 Sample Collection

Over 34 days in August-September 2015, PM$_{2.5}$ samples were collected from a kitchen in the village of Khatela, Palwal, Haryana, India. Figure 1 shows (A) the kitchen setup and (B) the stoves (*angithi* and *chulha*) and fuels (dung and brushwood) used. The stoves and fuels were obtained locally and traditional meals were prepared by a local cook. The cook was instructed by the experimenters to prepare a particular, standard meal using the selected fuel and stove. All *angithi* cookstoves burned dung and were used to prepare buffalo fodder. *Chulha* cookstoves burned either brushwood or dung fuels

and were used to prepare a traditional meal of *chapati* and vegetables for 4 people. Vegetables were cooked in a pressure cooker that rests on top of the *chulha* (Figure 1B). *Chapatti* were cooked in the air space next to the fuel, as is typical for this area. Brushwood/*angithi* cookfires were never tested because this combination is not frequently used in the local households. PM$_{2.5}$ emissions were sampled via three-pronged probes that hung above the cookstove. Air sampling pumps (PCXR-8, SKC Inc.) created a flow of BBOA emissions through aluminum tubing during cooking events. PM$_{2.5}$ was captured through

cyclone fractionators (2.5 µm cut point, URG Corporation) and the resultant flow was taken through a stainless steel filter holder containing PTFE filters (SKC Inc., 47 mm). One filter was collected for chemical analysis, and another filter for gravimetric analysis. Flows were measured via a mass flowmeter (TSI 4140) before and after each cooking event to ensure it had not varied more than 10%. The pump was turned on before cooking began so that emissions from the entire cooking event were captured and turned off when the fire was out. Prior to analysis, filters were stored at -80°C other than during

transportation and use. This includes time at the field site (1-6 hours) and transportation back to the United States (24 hours).



## 2.3 Nano-DESI-HRMS analysis

$PM_{2.5}$ collected on PTFE filters were analyzed with an LTQ-Orbitrap[TM] high resolution mass spectrometer (ThermoFisher Scientific) equipped with a custom built nano-DESI source (Roach et al., 2010a, 2010b). The solvent mixture (70% $CH_3CN$/30% $H_2O$) flowed through an electrified capillary at a flow rate of 0.3-1 µL/min, and extracted $PM_{2.5}$ in a small (<1

5  mm) droplet moving across the filter's surface at roughly 0.2 cm/min. The extract then flowed through the nanospray capillary, and into the mass spectrometer inlet. The spray voltage was 3.5 kV; the instrument was operated in positive ion mode. The instrument was calibrated with a standard mixture of caffeine, MRFA, and Ultramark 1621 (ThermoFisher Scientific). Two separate mass spectra were obtained from different portions of the filter to ensure reproducibility. Only peaks that showed up in both spectra were retained for further analysis.

Peaks with signal-to-noise ratios of greater than 3 were extracted from the time-integrated nano-DESI chromatograms using Decon2LS software. Peaks containing [13]C isotopes were excluded from analysis. Sample and solvent blank mass spectra peaks were clustered with a tolerance of 0.001 $m/z$ using a second-order Kendrick analysis with $CH_2$ and $H_2$ base units (Roach et al., 2011). The spectra were internally calibrated by assigning prominent peaks of common BBOA compounds first, and fitting the observed-exact $m/z$ deviation to a linear regression curve. The $m/z$ correction introduced by the internal

calibration was <0.001 $m/z$ units, but even at these small levels, the correction helped reduce the ambiguity in the assignments of unknown peaks. We focused on analyzing peaks with $m/z$< 350, as peaks above this $m/z$ value were small in abundance and in many cases could not be assigned unambiguously. Exact masses were assigned using the freeware program Formula Calculator v1.1 (http://magnet.fsu.edu/~midas/download.html). The permitted elements and their maximal numbers of atoms were as follows: C (40), H (80), O (35), N (5), and Na (1). Peaks that could not be assigned within the described

parameters had small abundances and were not pursued further. There were a few notable exceptions, namely, the potassium salt peaks discussed below. The double-bond equivalent (DBE) values of the neutral formulas were calculated using the equation: DBE = C - H/2 + N/2 + 1, where C, H, O, and N correspond to the number of carbon, hydrogen, oxygen, and nitrogen atoms, respectively.

## 2.4 HPLC-PDA-HRMS

The samples were further analyzed with an HPLC-PDA-HRMS platform (Lin et al., 2016). To prepare the samples for analysis, half of the PTFE filter was extracted overnight in mixture of acetonitrile, dichloromethane, and hexane solvents (2:2:1 by volume, 5 mL total), which was empirically found to work well for extracting a broad range of BBOA compounds (Lin et al., 2017). The solutions were then filtered with PVDF filter syringes to remove insoluble particles (Millipore, Duropore, 13mm, 0.22 µm). The solutions were concentrated under $N_2$ flow, and then diluted with water and dimethyl

sulfoxide (DMSO) to a final volume around 150 µL. The separation was performed on a reverse-phase column (Luna C18, 2 x 150 mm, 5 µm particles, 100 Å pores, Phenomenex, Inc.). The mobile phase comprised of 0.05% formic acid in LC/MS grade acetonitrile (B) and 0.05% formic acid in LC/MS grade water (A). Gradient elution was performed by the A/B mixture



at a flow rate of 200 µL/min: from 0-62 min hold at 90% A, 63-89 min hold at 10% A, 90- 100 min hold at 0% A, then 101-120 min hold at 90% A. The ESI settings were as follows: 5 µL injection volume, 4.0 kV spray potential, 35 units of sheath gas flow, 10 units of auxiliary gas flow, and 8 units of sweep gas flow. The solutions were analyzed in both positive and negative ion ESI/HRMS modes.

The HPLC-PDA-HRMS data were acquired and first analyzed using Xcalibur 2.4 software (Thermo Scientific). Possible exact masses were identified by LC retention time using the open source software toolbox MZmine version 2.23 (http://mzmine.github.io/) (Pluskal et al., 2010). Formula assignments were obtained from their exact $m/z$ values using the Formula Calculator v1.1

## 2.5 MAC and AAE

Selected filter halves of the samples were extracted as described in section 2.4. Absorption spectra were collected with a dual-beam UV-Vis spectrophotometer (Shimadzu UV-2450). Bulk mass absorption coefficient (MAC) values were calculated from the following equation:

$$MAC(\lambda) = \frac{A_{10}(\lambda) \cdot \ln(10)}{b \cdot C_{mass}}$$
(1)

where $A_{10}$ is the base-10 absorbance, b is the path length (cm), and $C_{mass}$ is the solution mass concentration in (g cm$^{-3}$). The largest uncertainty in MAC came from uncertainty in $C_{mass}$ of the extract. First, the overall mass of PM$_{2.5}$ on the filter had to be estimated from another filter collected specifically for gravimetric analysis. The particle mass distribution on the filter was assumed to be uniform, and the maximum extraction efficiency was estimated to be 50% by weighing a temperature and

relative humidity conditioned filter before and after the extraction. Since the in many cases the extraction efficiency was lower than this, the MAC values reported here represent lower limits. Absorption angstrom exponents (AAE) were calculated for both samples by fitting the log(MAC) vs. log($\lambda$) to a linear function over the wavelength range of 300 to 700 nm.

## 3 Results and Discussion

### 3.1 Nano-DESI-HRMS analysis of cookstove particles

Representative nano-DESI mass spectra from the three major types of cookfires sampled are shown in Figure 2. Approximate emission factors (EFs) were estimated assuming that the peak abundances are proportional to the mass concentrations of the observed species (see SI section for details). We want to emphasize that the EFs calculated this way are approximate, and should be used as upper limits and orders of magnitude estimates. It is clear from the mass spectra in

Figure 2 that the 3 combinations of fuel/stove types lead to distinct particle compositions.



We compare the particle composition of the three major cookfire types by averaging the percentage of $C_xH_yN_w$, $C_xH_yO_z$, and $C_xH_yO_zN_w$ peaks in the nano-DESI spectra from multiple samples. Samples used and a summary of the following discussion is detailed in Table S1.1. The overwhelming majority of detected species by nano-DESI in dung cookfire smoke $PM_{2.5}$ was attributed to $C_xH_yN_w$, compounds that contain only carbon, hydrogen, and nitrogen atoms. The average count-based fractions

from $C_xH_yN_w$ species were 79.9%±4.4% and 82.1%±1.0% for dung/*chulha* and dung/*angithi* experiments, respectively, but only 23.8%±7.8% for brushwood/*chulha* experiments. This was somewhat unexpected since all nitrogen-containing compounds in the smoke $PM_{2.5}$ should reflect nitrogen content of the fuels (Coggon et al., 2016) which was roughly the same (1.4±0.3 and 1.4±0.1 for brushwood and dung, respectively) (Gautam et al., 2016). On the other hand, $PM_{2.5}$ from brushwood cookfire smoke contained higher fractions of $C_xH_yO_z$ species: 43.1%±14.6% in brushwood/*chulha* cookfires

were assigned as $C_xH_yO_z$ species compared to only 4.1%±0.9% and 3.2%±3.3% for dung/*chulha* and dung/*angithi* experiments, respectively. Many of the $C_xH_yO_z$ formulas were consistent with species reported previously as lignin pyrolysis products. Fractions of $C_xH_yO_zN_w$ did not correlate well with fuel/stove variables and ranged from 4.1% to 34.4% in the analyzed samples.

Inorganic salt peaks containing potassium and chlorine were observed in more than half of dung cookfires (8 out of 14) and

15 all brushwood cookfires. These peaks were pursued apart from the original analysis because the peak abundance was very large in many mass spectra. These mass spectra all contained $K_2Cl^+$ as the most prominent salt peak, and $K_3Cl_2^+$ was also present in a few mass spectra. Isotopic variants of these salts, namely with either $^{37}Cl$ or $^{41}K$ (24% or 6.7% natural abundance) instead of $^{35}Cl$ or $^{39}K$ (76% or 93.3% natural abundance), were also found. The resolving power of the HRMS instrument is insufficient to distinguish the isotopic shifts from Cl and K ($\Delta$ mass$_{37Cl-35Cl}$= 1.997 Da, $\Delta$ mass$_{41K-39K}$= 1.998

20 Da) but one or both of the isotopes were consistently present in all mass spectra containing potassium ions. Adducts corresponding to a replacement of K by Na were also detected. The main source of potassium may have been not the biomass itself but rather the result of the food items cooked or the stove material itself. Inorganic salts were observed in all *chulha* cookfire $PM_{2.5}$ samples regardless of fuel type and were absent in all *angithi* cookfire $PM_{2.5}$ samples. These stoves produced meals for people or animal fodder, respectively. The *chulha* was made mainly from brick with a local covering of

25 local clay, whereas the *angithi* only from clay. With the presently available data it is impossible to determine whether the potassium salts originated from the *chulha* material or is the result of different food items cooked.

Levoglucosan, a commonly used biomass burning tracer (Simoneit et al., 1999), was present in 3 out of 8 dung/*chulha* cookfires, 4 out of 6 dung/*angithi* cookfires, and 4 out of 11 brushwood/*chulha* cookfires. Levoglucosan appears to be just as good a tracer for digested biomass burning as for woody biomass burning.

**3.2 Particle-phase biomass burning tracers**

An inventory of compounds that were reproducibly observed in samples from three different cooking events using the same fuel/stove combinations was compiled. The peak abundances were first normalized to the largest peak abundance then the



three mass spectra were averaged. Peaks that did not appear in all three mass spectra were discarded. Since the absolute peak abundances varied in individual spectra, only approximate relative abundances are reported here grouped into three logarithmic bins, denoted as LOW (<1%), MEDIUM (1-9.99%), HIGH (10-100%). This analysis was completed for the emissions from each of the three types of cookstove-fuel combinations studied in this work. Fuel moisture contents were also

considered with the goal of comparing emissions from fuel with similar dryness (see SI Table S1.1 for details).

Figure 3 summarizes how reproducibly-detected PM$_{2.5}$ compounds are organized in the inventory. First, we will provide a list of compounds common to the emissions from all 3 types of cookfires including: dung/*chulha*, dung/*angithi*, and brushwood/*chulha* (Section 3.3, Table 1). Then, we will discuss compounds exclusively found in the brushwood/*chulha* cookfire emissions (Section 3.4, Table S3.1). We next show compounds common to dung cookfire emissions (Section 3.5.1,

Table 2). Lastly, we discuss BBOA compounds detected in all dung cookfire experiments (Section 3.5.2). Within section 3.5.2 we discuss compounds unique to the dung/*chulha* (Table S3.2) and the dung/*angithi* (Table S3.3) cookfire experiments, as well as the compounds they had in common (Section 3.5.1, Table 2).

The numbers of reproducibly-detected formulas are shown in Figure 3 in blue. We found that the chemical composition of PM$_{2.5}$ from dung cookfires was far more complex (i.e., had more observed peaks) than PM$_{2.5}$ from brushwood cookfires.

Further, the PM$_{2.5}$ from dung/*angithi* cookfires was more complex than dung/*chulha* cookfires. There were 93 compounds reproducibly detected in the brushwood/*chulha* cookfire PM$_{2.5}$ samples compared to 212 and 262 for dung/*chulha* and dung/*angithi* cookfires, respectively. There were five compounds the *chulha* cookfires had in common, with two of them being the potassium salt peaks described earlier. There was one compound (C$_{14}$H$_{16}$O$_3$) shared by only dung/*angithi* and brushwood/*chulha*. Because of the small number of these peaks, they will not be discussed in this paper. In the following

sections, we will discuss compounds that are common in all cookfires, as well as unique compounds.

Figure 4 summarizes the BBOA inventory described in more detail in sections 3.3-3.5, i.e., compounds common to dung/chulha, dung/angithi, brushwood/chulha cookfires; compounds found exclusively in the emissions from brushwood/chulha cookfires; and species that are unique to dung cookfires. Figure 4A pie charts compare the fraction of count-based, normalized abundance in each elemental category. PM$_{2.5}$ compounds shared among all samples of this study are

diverse. In terms of count-based abundance, compounds emitted from all dung-burning cookfires are largely nitrogen-containing. From Figure 4B, the common compounds make up the vast majority (97%) of detected compounds from the brushwood/*chulha* cookfires. Similarly for the dung cookfires, the common cookfire compounds (grey) and dung cookfire compounds (brown) make up 95% or more of the mass spectra abundance as shown in Figure 4B. Therefore, the common compounds (Table 1) and dung compounds inventories (Table 2) contain the bulk of the PM$_{2.5}$ species in terms of count-

based abundance.

### 3.3 Compounds common to dung/*chulha*, dung/*angithi*, brushwood/*chulha* cookfires



Table 1 provides a complete list of eighty reproducibly-detected compounds that were common to emissions from all cookfires. These common compounds make a large contribution to the mass spectra for every cookfire type (Figure 4), with MEDIUM being the most common relative abundance given in Table 1. More than half of the abundance (59%) was due to the nitrogen-containing compounds ($C_xH_yN_w$ or $C_xH_yO_zN_w$), as shown in Figure 4a. ESI detection likely biases the elemental

make-up of smoke $PM_{2.5}$, as nitrogen-containing species are more easily ionized compared to sugars and lignin (Wan and Yu, 2006). Nevertheless, the brushwood and dung fuels in Gautam et al., (2016) have similar nitrogen contents, and a large overlap in the $C_xH_yN_w$ and $C_xH_yO_zN_w$ species was observed. Fuels used by Gautam et al., (2016) were collected in the same area and around the same time as the fuels used in this study.

The common compounds make up a large fraction for all cookfire types. This is especially true for the sample from

brushwood/*chulha* cookfires, where their fraction is ~86% in number. Many of these $C_xH_yO_z$ species have elemental formulas consistent with typical lignin- and sugar-derived products such as anisaldehyde, veratraldehyde, vinylguaiacol, syringylethanone, trimethoxyphenylethanone, etc. reported previously in the literature (Laskin et al., 2009; Simoneit et al., 1993; Smith et al., 2009). These tentative molecular assignments are listed in Table 1 alongside their elemental formulas. Approximately 20% of the common compounds (17 out of 80 formulas) have been also identified in earlier studies reporting

molecular characterization of $PM_{2.5}$ samples collected from burning of one or more of the following fuels: Alaskan duff, ponderosa pine duff, southern United States pine needles, or ceanothus fuels (Laskin et al., 2009; Smith et al., 2009). Many of these fuels are non-woody and all are undigested biomass, very different kinds of biomass from those used as cookstove fuels in this study and in this region of India. This suggests that at perhaps 20% of the compounds listed in Table 1 might be commonly detected in BBOA samples, regardless of biomass type.

**3.4 Compounds found exclusively in the emissions from brushwood/*chulha* cookfires**

Table S3.1 lists the compounds found exclusively in the samples from brushwood/*chulha* cookfires. Many of them correspond to lignin-derived products that have been previously identified in BBOA by gas chromatography methods, as indicated in Table S3.1 (Lee et al., 2005; Simoneit, 2002; Simoneit et al., 1993; Smith et al., 2009). Lignin is an essential component of wood, comprising roughly a third of its dry mass (Collard and Blin, 2014; Simoneit, 2002). Lignin is generally

composed of *p*-coumaryl, confieryl, and syringyl alcohol units. During pyrolysis, the coumaryl, vanillyl, and syringyl moieties, respectively, are preserved and are found in smoke. More generally, the lignin pyrolysis products found in smoke contain a benzene ring, often with hydroxy and/or methoxy substituents. Based on these previous observations and the assumption that these are lignin pyrolysis products, tentative molecular structures were assigned to $C_xH_yO_z$ compounds. It is likely that some $C_xH_yO_z$ molecular species specific to the emissions from the brushwood burning were not detected in this

study due to their low ionization efficiency.

**3.5 Species Unique to Dung Smoke $PM_{2.5}$**

**3.5.1 Compounds emitted from both dung/*angithi* and dung/*chulha* cookfires**





Overall, chemical composition of $PM_{2.5}$ samples of dung-burning emissions is far more complex than the samples from the brushwood-burning cookfires. Table 2 lists the 115 compounds found exclusively and reproducibly in the dung-fueled samples. These compounds are largely $C_xH_yN_w$, as shown in Figure 4b. Only a few of the elemental formulas, $C_8H_{16}N_2$, $C_{11}H_8N_2$, and $C_{13}H_{11}ON$, have been reported previously (Laskin et al., 2009; Smith et al., 2009).

**3.5.2 Analysis of compounds found in all dung-burning cookfires**

In addition to the common dung compounds listed in Table 2, there were compounds detected exclusively in the emissions from either dung/*chulha* cookfires (Table S3.2) or dung/*angithi* cookfires (Table S3.3). All of these compounds are nitrogen-containing, and none have been reported previously, to the best of our knowledge. Hereafter in this section, we combine all compounds found in dung-burning cookfire $PM_{2.5}$, presented in Tables 2, S3.2, and S3.3, and discuss their possible

molecular character.

Figure 5 shows the double bond equivalent (DBE) as a function of the carbon number of compounds detected in all investigated samples. The DBE versus C dependence for classes of compounds with different degrees of unsaturation, including: terpenes (red), polyenes (orange), polycyclic aromatic hydrocarbons (yellow shaded, PAHs) are also shown to aid the classification of the compounds observed in the $PM_{2.5}$ samples. Thirty of the 193 formulas fall in the PAH region of the

plot suggesting that they have aromatic structures (Figure 5a). Figure 5b compares the DBE values of the molecular components detected in the emissions exclusive to brushwood/*chulha* cookfires (Table S3.1) and the common compounds from all studied samples (Table 1). In general, the DBE increases with carbon number for the compounds common to all cookfires. Only eight of the 87 compounds fall directly in the PAH region. There are more aromatic structures specific to the dung smoke compared to the compounds detected in all cookfires.

Detected nitrogen compounds with high DBE values are likely N-heterocyclic PAH compounds. Figure 6 displays possible structures for the select detected nitrogen-containing compounds with a high DBE. Purcell et al., (2007) found that pyridinic PAH compounds were readily ionized from standard mixtures of N-heterocyclics in positive-ion ESI. This gives us more confidence in our observation of $C_{13}H_9N$, tentatively acridine, and $C_{11}H_8N_2$, tentatively β-carboline, which have pyridinic nitrogen atoms and likely have high ionization efficiencies. The peak abundances of these compounds are significant, with

medium and high designations, respectively. $C_{12}H_9ON$ cannot have a pyridinic nitrogen and is tentatively assigned as phenoxazine.

Kendrick analysis identifies homologous series of structurally related compounds that share a core formula and differ in the number (n) of additional $CH_2$ units (Hughey et al., 2001). 172 of the 193 detected compounds from the dung-burning cook fire emissions can be grouped into 43 homologous series based on the Kendrick mass defect plot, as shown in Figure 7.

There are 15 homologous series and 5 independent formulas that make up the 61 total $C_xH_yN_w$ (red) compounds. This suggests that there are at least 20 distinct types of structures that made up the observed $C_xH_yN_w$ species. Similarly, there are 30 homologous series for $C_xH_yO_zN_w$ (purple) formulas and 12 $C_xH_yO_zN_w$ formulas yielding at least 42 distinct types of





structures for this formula category. There are no homologous series from $C_xH_yO_z$ species, presumably because only a few members of this group can be detected by ESI-based methods in the $PM_{2.5}$ from the dung cookfires. From this analysis, we find that on the whole, there are at least 66 unique types of structures from the 193 compounds detected from dung-burning cookfire emissions. This Kendrick analysis suggests that some of the observed N-heterocyclic PAHs have alkyl substituents.

5 For example, phenoxazine and β-carboline (Figure 6) serve as the core molecules in the homologous series $C_nH_{2n-15}ON$ and $C_nH_{2n-14}N_2$, respectively (Figure 7).

**3.6 Light-absorbing properties and chromophores from cookstove emissions**

MAC values determined for the solvent extractable components of the samples are shown in Figure 8. MAC values for the samples from the brushwood burning are roughly twice the MAC values of dung between 300-580 nm. However, $PM_{2.5}$

10 emission factors (a detailed analysis of the emission factors will be reported in a follow up paper) are a factor of 2.5 higher for dung/*chulha* fires (12.3±2.5 g kg$^{-1}$ fuel) compared to brushwood/*chulha* fires (4.9±1.7 g kg$^{-1}$ fuel). Taken together, the MAC values and particle EFs indicate that the overall absorption coefficient by the BBOA is likely comparable for dung and brushwood/*chulha* emissions.

MAC values at 400 nm were 1.9 m$^2$ g$^{-1}$ and 0.9 m$^2$ g$^{-1}$ for the samples from brushwood/*chulha* and dung/*chulha* cookfires,

15 respectively. Kirchstetter and colleagues reported MAC of 2.9 m$^2$ g$^{-1}$ at 400 nm for the BrC in biomass smoke samples (Kirchstetter et al., 2004). Chen and Bond measured MAC values at 360 nm of nearly 2.0 m$^2$ g$^{-1}$ for methanol extracts of particles resulting from oak pyrolysis, and nearly 2.5 m$^2$ g$^{-1}$ for pine wood pyrolysis (Chen and Bond, 2010). Our MAC value at 360 nm for brushwood was larger at 3.7 m$^2$ g$^{-1}$, possibly due to a more efficient extraction of a broader range of chromophores by the solvents utilized. The pyrolysis temperature and wood composition could also contribute to the

20 difference. Our MAC value at 360 nm for dung was lower compared to our brushwood sample at 1.9 m$^2$ g$^{-1}$. This could be a combined result of the likely lower pyrolysis temperature and difference in the biomass composition (Chen and Bond, 2010). The AAE values for the brushwood and dung samples are 7.5 and 6.8, respectively. Our brushwood AAE fits into the lower end of the AAE range presented in Chen and Bond, 6.9 to 11.4 (Chen and Bond, 2010). However, the observed absorption can be definitively attributed to BrC since AAE values of 2 or greater indicate that light absorption comes from BrC as

25 opposed to BC (Kirchstetter et al., 2004; Laskin et al., 2015). Typical AAE values cited in the literature for BrC in BBOA are in a range of 2-11 (Kirchstetter et al., 2004; Laskin et al., 2015).

We now focus on identifying the main chromophores that contribute to the high MAC we observe for cookstove $PM_{2.5}$. Two cookfires using dung and brushwood fuels were selected for a more detailed analysis of the light-absorbing molecules (BrC chromophores). The dung cookfire utilized an *angithi* cookstove to prepare buffalo food. The brushwood cookfire was used

30 to prepare a traditional meal of rice and lentils with a *chulha*. More detailed sample information is provided in Table S1.3. The samples were analyzed using HPLC-PDA-ESI/HRMS platform following the methods described elsehwere (Lin et al., 2015, 2016, 2017). The identified chromophores and their PDA chromatograms are illustrated in Figure 9, and the retention





times and peaks in the absorption spectra are listed in Tables 3 and 4 for the emissions from brushwood and dung cookfires, respectively.

The BrC chromophores for both brushwood and dung samples are largely $C_xH_yO_z$ compounds (Tables 4 and 5). We conclude that lignin-like BrC chromophores account for the majority of the extracted light-absorbing compounds in both samples. We

also found a few nitrogen-containing BrC chromophores (e.g., $C_9H_7NO_2$ and $C_8H_9NO_3$) in both the brushwood and dung samples. The woody and digested biomasses shared 3 strongly-absorbing chromophores, $C_8H_8O_4$ (tentatively vanillic acid), $C_{10}H_{12}O_3$ (tentatively ethyl methoxybenzoate), and $C_{13}H_{10}O_2$, as well as comparably weaker absorbing chromophores. $C_{10}H_{10}O_3$ is another strong absorber of near UV radiation that was found in both samples. In the brushwood-derived $PM_{2.5}$, $C_{10}H_{10}O_3$ elutes at 18.32 min ($\lambda_{max}$= 337 nm), while in the dung smoke sample, it is not oberved until 24.54 min ($\lambda_{max}$= 299,

308 nm). These are clearly different chromophores with the same chemical formula, possibly coniferaldehyde and methoxycinnamic acid. $C_9H_8O_3$ is a similar case, in which the same chemical formula appears at different retention times in the selected ion chromatograms (SICs) for brushwood- and dung-derived $PM_{2.5}$. In the brushwood-derived $PM_{2.5}$ sample, $C_9H_8O_3$ coelutes with $C_9H_7NO_2$ at 17.25 min (Table 3). In the dung $PM_{2.5}$ sample $C_9H_8O_3$ coelutes with $C_8H_8O_4$ and $C_9H_{10}O_4$ at 14.44 min (Table 4). The $C_9H_8O_3$ formula could correspond to coumaric acid for either retention time. Because the

compound coelutes with other potential chromophores, we refrained from assigning a proposed structure to the chemical formula.

There were light-absorbing molecules specific to brushwood-derived $PM_{2.5}$ (Table 3) that could account for higher MAC values compared to the dung-derived $PM_{2.5}$. $C_8H_9NO_4$ is a possible nitroaromatic compound with its absorbance peaking around 335 nm. $C_8H_{10}O_3$, tentatively syringol, is closely related to syringic acid, a lignin monomer. The formula was also

detected in the dung-derived $PM_{2.5}$ sample, but the absorption was lower by approximately a factor of 20, and therefore is not considered a main chromophore.

There were strongly-absorbing BrC chromophores in the $PM_{2.5}$ generated by burning dung fuel that eluted in the first couple of minutes of the sample run (See Figure 9b). These early-eluting chromophores were likley polar compounds that were not retained well by the column and thus ignored in this analysis. For both $PM_{2.5}$ samples, most of the chromophores eluted in

the first 30 minutes of the run shown in Figure 9. Compounds eluting in the range of 30 to 60 min were also satisfactorily separated, but these were weakly absorbing. The non-polar PAH compounds absorbing in UV-Vis range are not ionized by the ESI source and subsequently not detected by HRMS (Lin et al., 2016). It is possible that additional light-absorbing molecules essential to dung smoke were strongly retained by the column and eluted after 60 min.

Absorption spectra recorded in tandem with the mass spectra provides additional constraints on the assignments. For

example, at 15.57 minutes, $C_{10}H_{12}O_3$ and $C_9H_{10}O_3$ coeluted in both BBOA samples. These compounds were given the tenative assignments of ethyl-3-methoxybenzoate and veratraldehyde, respectively. The UV-Vis absorbance of ethyl-3-methoxybenzoate shown in Figure 10 provides a reasonable match for the recorded PDA spectra for both samples at a retention time of 15.57 min. Veratraldehyde, which is derived broadly from lignin, has an absoprtion spectrum peaks at 308



nm in aqueous solution (Anastasio et al., 1997). Therefore, both ethyl-3-methoxybenzoate and veratraldehyde contribute to the spectrum observed by the PDA detector, and cannot be completely separated with this HPLC protocol.

For many formulas, multiple structural isomers were observed in SICs, with peaks appearing at more than one retention time. This behavior has been observed for other types of BBOA samples, described in Lin et al., 2016, and is inherent to lignin's

nature, such that a single $C_xH_yO_z$ chemical formula can correspond to multiple possible structural isomers. There are several cases in which chemical formulas show up multiple times in Tables 4-5. An example from the brushwood $PM_{2.5}$ (Table 3) is $C_9H_{10}O_4$ which elutes at 10.57 and 14.44 minutes. $C_9H_{10}O_4$ has been previously found in lignin pyrolysis BBOA in the forms of homovanillic acid and syringealdehyde (Simoneit et al., 1993). $C_8H_8O_4$ and $C_9H_{10}O_3$ are additional examples of the similar occurance in the sample of dung-derived $PM_{2.5}$, as they both appear twice in the SICs as shown in Table 4. One peak

corresponding to $C_8H_8O_4$ is very likely vanillic acid (Simoneit, 2002; Simoneit et al., 1993). $C_9H_{10}O_3$ could be either veratraledehyde or homoanisic acid, both have been observed from lignin pyrolysis (Simoneit et al., 1993). Collectively, these results indicate that many of the lignin-like chromophores have multiple structural isomers, some of which have likely been observed before (Simoneit, 2002; Simoneit et al., 1993).

## 4 Summary

Molecular analysis of $PM_{2.5}$ emissions from the three types of cookstove-fuel combinations showed that the chemical complexity of particle composition increased in the following order: brushwood/*chulha*, dung/*chulha*, dung/*angithi*. The compounds accounting for the additional complexity in dung-derived emissions were mostly $C_xH_yO_zN_w$ and $C_xH_yN_w$ species, which have not been identified before in BBOA. A substantial portion of the compounds specific to dung cookfires were aromatic. The $CH_2$-Kendrick analysis of the nitrogen-containing species from dung cookfires indicated that many may be

structurally related by substitution with alkyl chains of variable length.

The estimated MAC values for the $PM_{2.5}$ emissions samples from brushwood/*chulha* and dung/*chulha* cookfires were comparable in magnitude and wavelength dependence to the values previously observed for BBOA samples. While the MAC values for the brushwood-derived BBOA were higher than those for the dung-derived BBOA, the particle emission factors had the opposite relationship. Therefore, per unit mass of burned fuel, the dung and brushwood fueled cookstoves may have

25 comparable contribution to the overall light absorption. A set of $PM_{2.5}$ samples from brushwood/*chulha* and dung/*chulha* cookfires was analyzed using HPLC-PDA-HRMS to identify BrC chromophores. The vast majority of chromophores observed for both fuel types were lignin-like $C_xH_yO_z$ compounds. There were 3 retention times at which strongly-absorbing chromophores eluted for both samples: $C_8H_8O_4$ (vanillic acid), $C_{10}H_{12}O_3$ (methoxybenzoate), and $C_{13}H_{10}O_2$. There were also fuel-specific chromophores such as $C_{10}H_{10}O_3$ (distinct isomers for each fuel type), $C_8H_{10}O_3$ (syringol, brushwood), and

30 $C_{12}H_{10}O_4$ (dung).

This study suggests that there are a wide range of particle-phase compounds produced by cookstoves, beyond the lignin-like $C_xH_yO_z$ compounds that have previously been identified in wood burning studies. Specifically, from dung cookfires, we detected what we presume to be aromatic nitrogen-containing compounds with few or no oxygen atoms in them. Our



inventory of chemical formula is just the starting point for comprehensively characterizing particle-phase cookstove emissions. Future efforts should focus on the identification of compounds, more precise emission factor quantification for specific compounds, evaluation of toxicity, and modeling the effect of these compounds on secondary air pollution formation in aging smoke plumes.

## 5 Acknowledgements

This research was supported by EPA STAR Grant R835425 Impacts of household sources on outdoor pollution at village and regional scales in India. The contents are solely the responsibility of the authors and do not necessarily represent the official views of the US EPA. US EPA does not endorse the purchase of any commercial products or services mentioned in the publication. The authors acknowledge NOAA grants NA16OAR4310101 (PNNL) and NA16OAR4310102 (UCI) for

supporting nano-DESI and HPLC-PDA-HRMS analysis of samples. The HRMS measurements were performed at the W.R. Wiley Environmental Molecular Sciences Laboratory (EMSL) – a national scientific user facility located at PNNL, and sponsored by the Office of Biological and Environmental Research of the U.S. DOE. PNNL is operated for U.S. DOE by Battelle Memorial Institute under Contract No. DE-AC06-76RL0 1830.

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



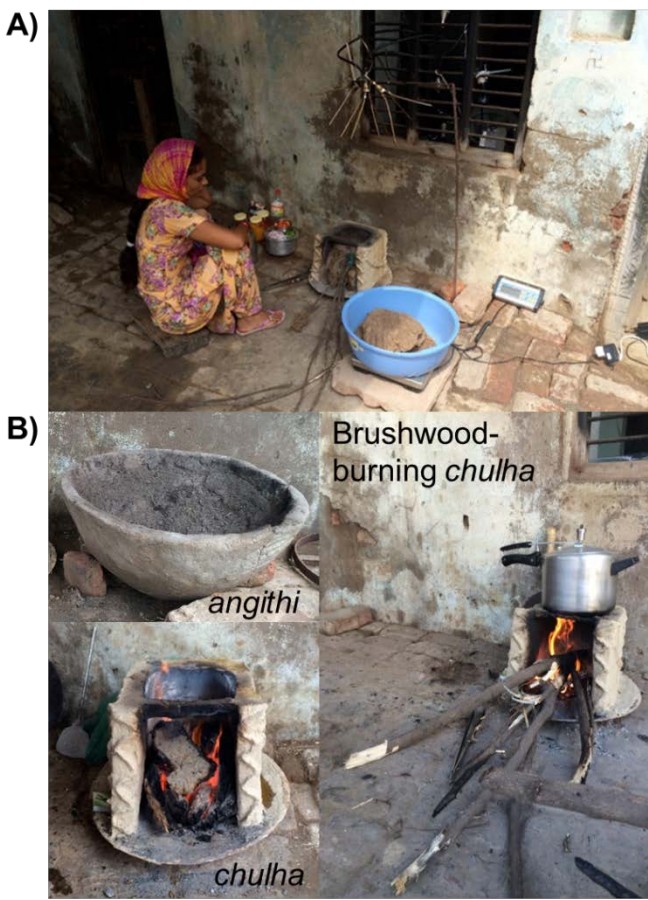

**Figure 1: The field site and set-up for cooking events. A) The kitchen set-up at the field site. B) The stoves and fuels used in this study:** *angithi***, dung-burning** *chulha***, and brushwood-burning** *chulha***.**





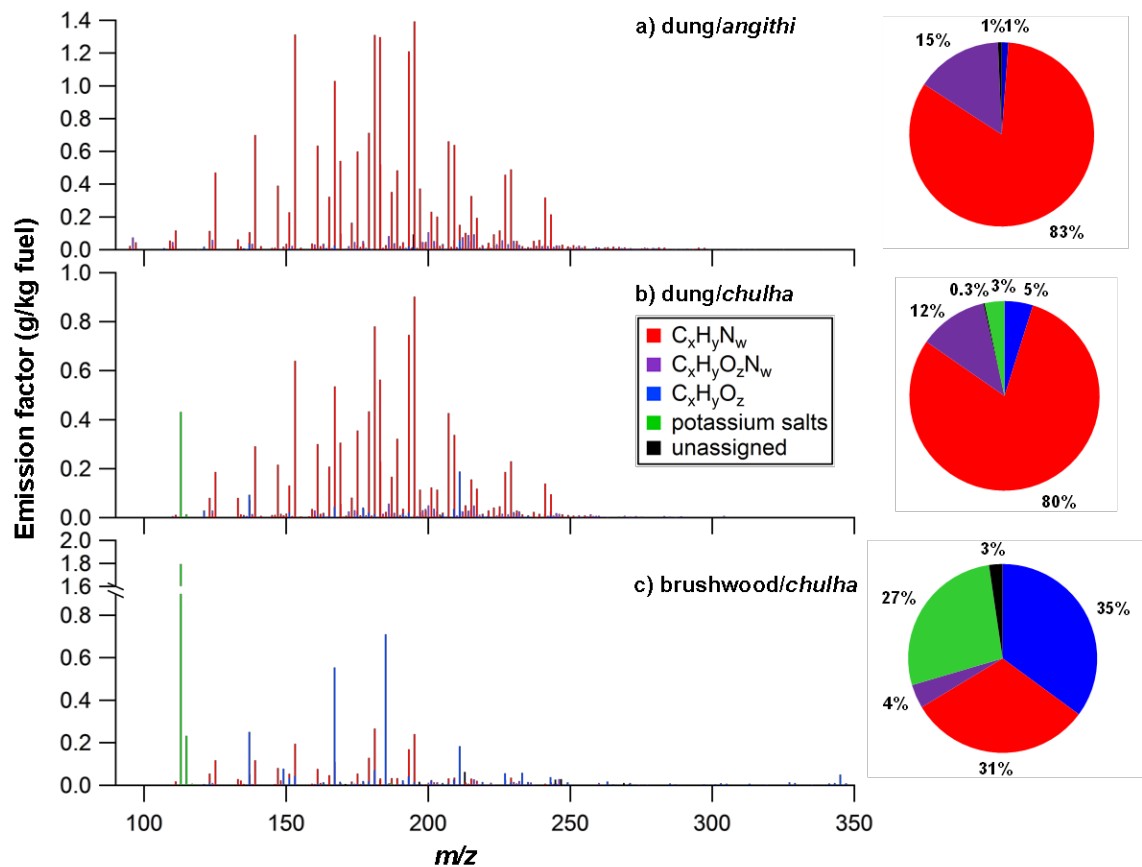

**Figure 2: Representative nano-DESI mass spectra collected for a) dung/*angithi* b) dung/*chulha* and c) brushwood/*chulha* cookfires. Approximate emission factor is plotted against *m/z*. Peaks are colored by their elemental makeup, $C_xH_yN_w$ (red), $C_xH_yO_zN_w$ (purple), $C_xH_yO_z$ (blue), potassium salts (green), and unassigned (black). The vertical axis represents approximate upper limits for the EFs (see SI for details). The pie charts illustrate the fraction of count-based, normalized peak abundance that is attributed to each elemental category.**





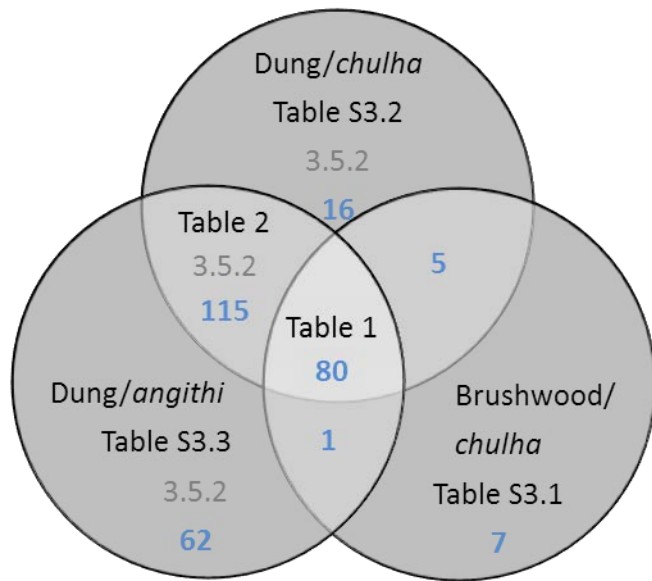

**Figure 3: An overview of the particle-phase compounds inventory based on the results of molecular characterization using nano-DESI-HRMS. Each area of the Venn diagram contains the bolded number of reproducibly-detected formulas in blue as well as the Table that lists peaks for each category. Merging all the Tables listed here provides a complete inventory of compounds detected in this study. Section 3.5.2 does not contain any tables and instead is a discussion of compounds in Tables 3, S3.2, and S3.3.**





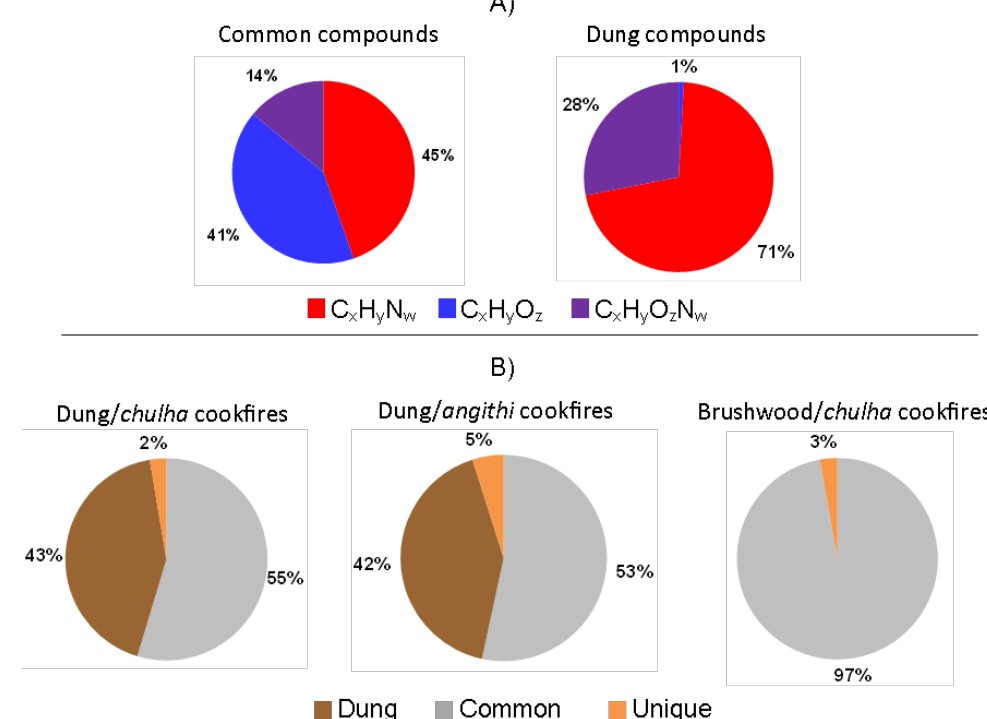

**Figure 4: A summary of the inventory in terms of the count-based, normalized peak abundances. A) Contribution of PM$_{2.5}$ compounds to each elemental formula category for those found in all cookfires and those found in all dung-burning cookfires. B) The compounds by cookstove type classified as compounds common to all cookfires in grey, compounds common to all dung cookfires in brown, and unique compounds in orange.**





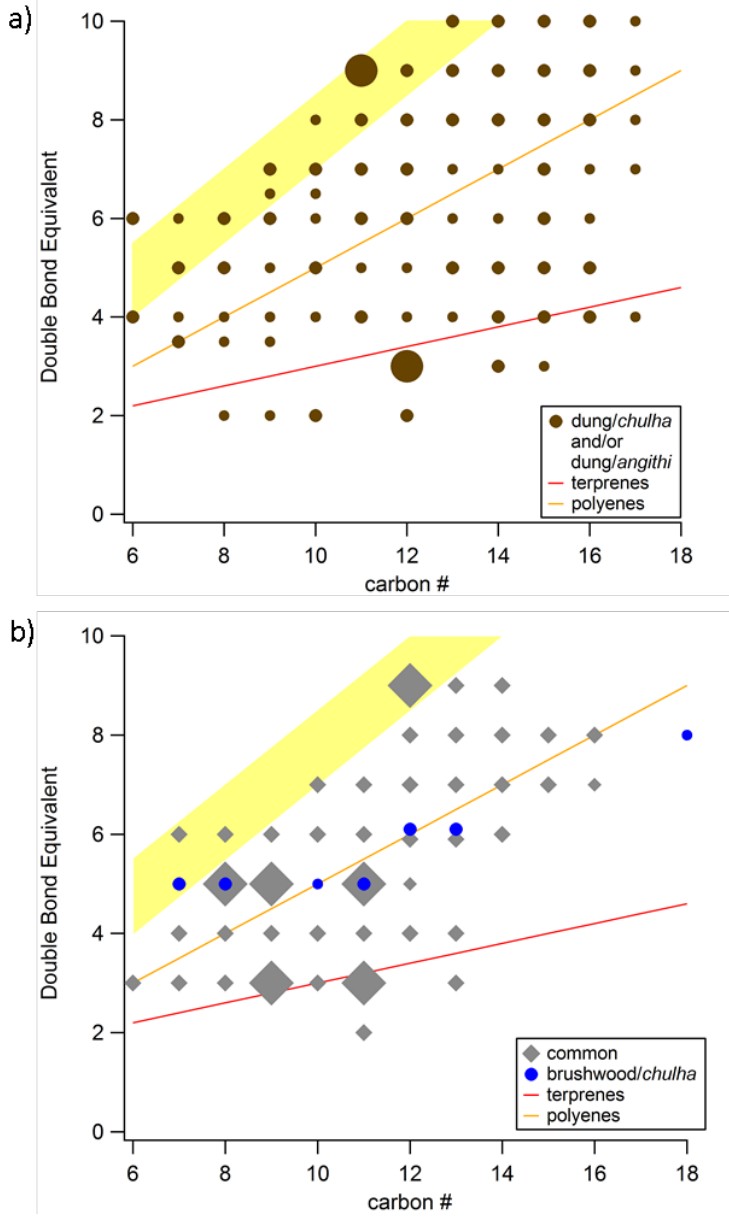

**Figure 5: Double bond equivalent (DBE) as a function of the carbon number for a) a combined set of compounds detected from all dung cookfires (brown circle) and b) compounds all cookfires have in common (grey diamond) as well as compounds exclusively found in brushwood (blue circle). Markers representing one or multiple species are sized by their LOW, MEDIUM, and HIGH designations. The curves illustrate theoretically where terpenes (red) and polyenes (orange) would fall. Similarly, the yellow-shaded region shows were PAHs would appear, including: cata-condensed PAHs with 0, 1, and 2 heterocyclic nitrogen atoms and circular PAHs.**





**Figure 6: Possible structures of N-heterocyclic PAHs found in dung cookfire emissions. C₁₃H₉N was detected reproducibly in dung/*chulha* emissions only, while C₁₂H₉ON and C₁₁H₈N₂ were reproducibly detected in all dung cookfires.**



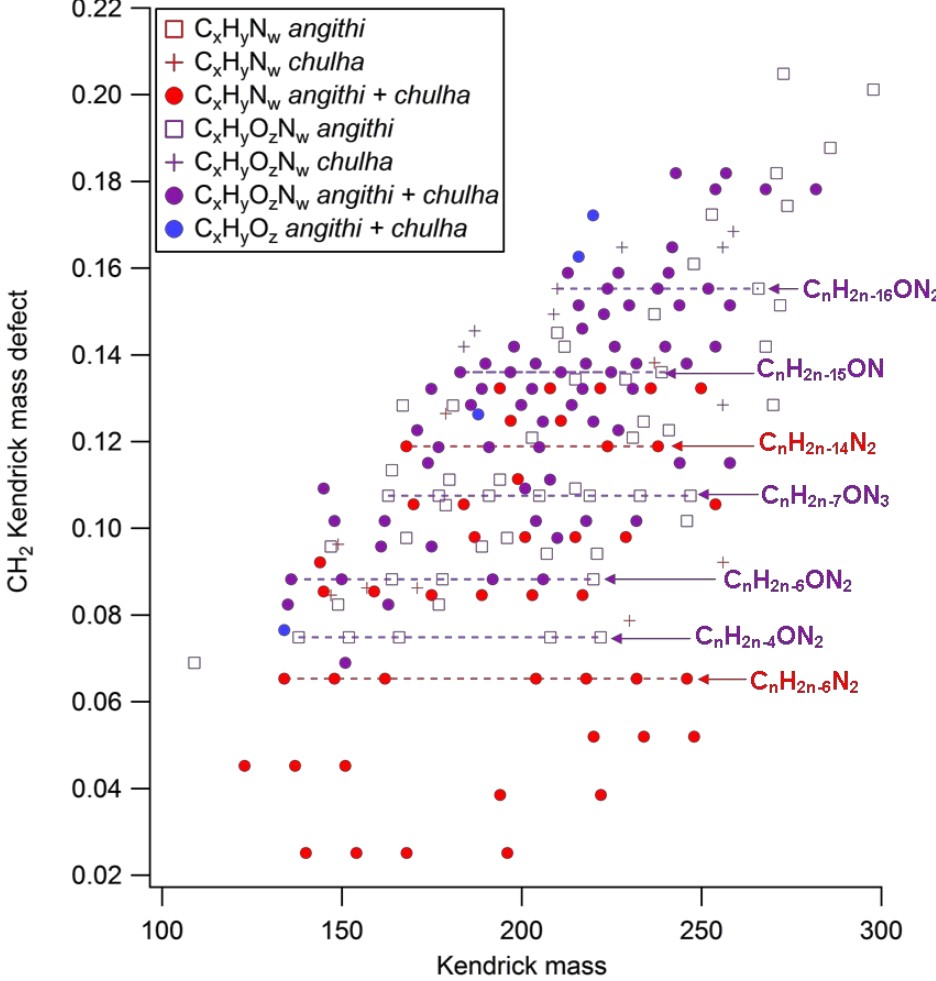

**Figure 7: The CH$_2$ Kendrick mass defect plot for compounds emitted only from dung stoves. The marker color determines the compound category for C$_x$H$_y$N$_w$ compounds (red), C$_x$H$_y$O$_z$ (blue), or C$_x$H$_y$O$_z$N$_w$ (purple). Marker shape indicates the stove(s) that reproducibly produced the compound: *chulha* and *angithi* (●), *angithi* (□), or *chulha* (+). Homologous series are identified with dotted horizontal lines, and suggest they have similar structures.**




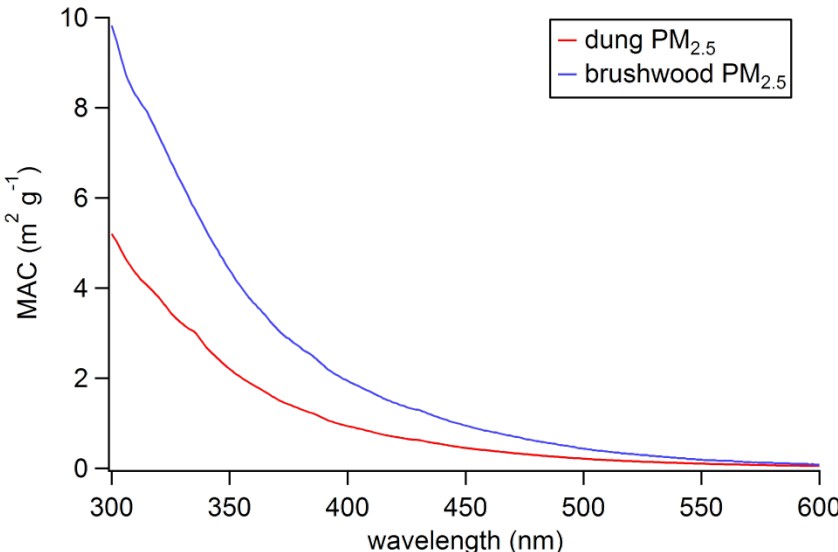

**Figure 8: Comparing MAC (m² g⁻¹) for brushwood /*chulha* (blue) and dung/*chulha* (red) samples.**



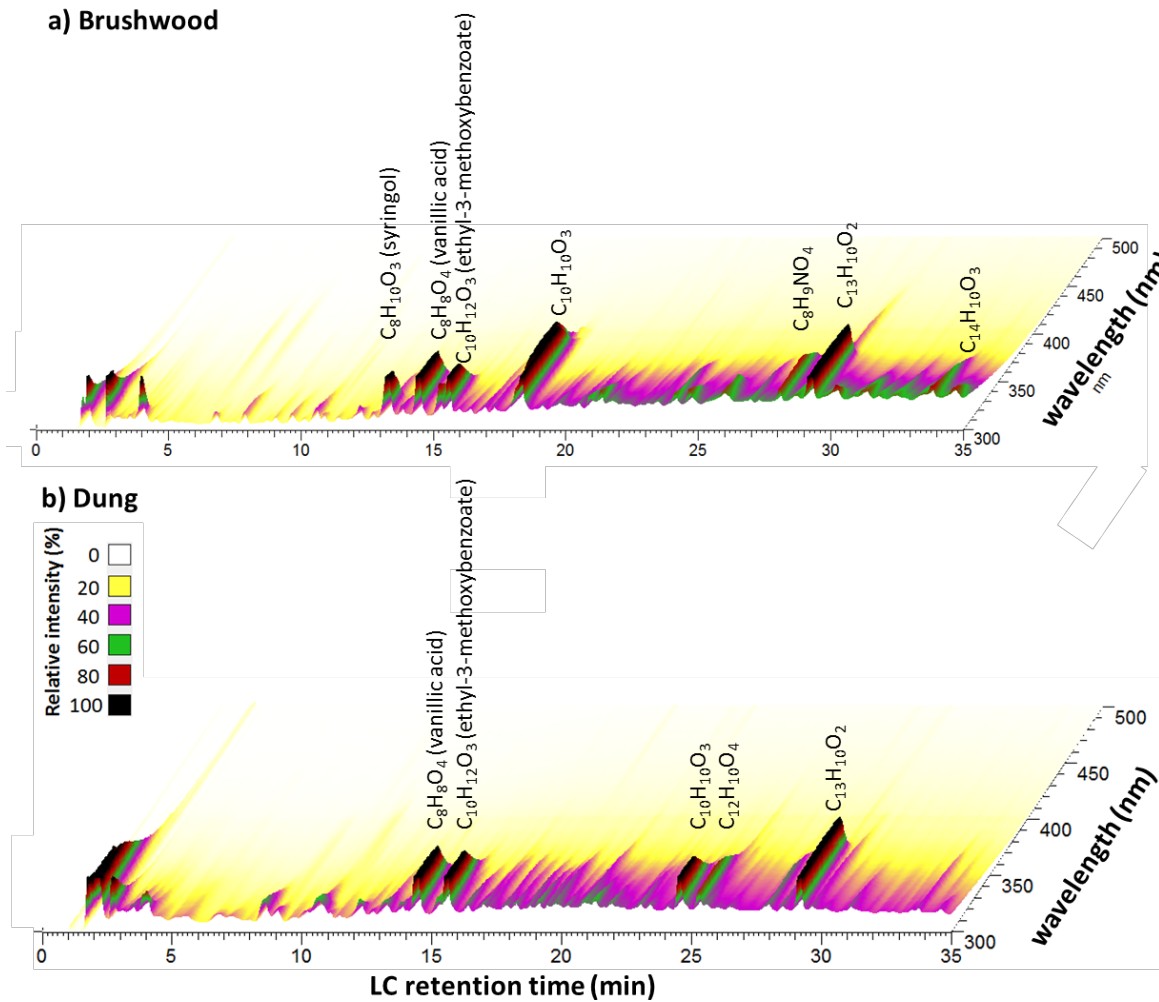

**Figure 9: HPLC-PDA chromatogram showing BrC chromophores detected in the emmision samples from a) brushwood and b) dung cookfires. The strongest-absorbing molecules and their corresponding PDA retention times are given above the peak.**



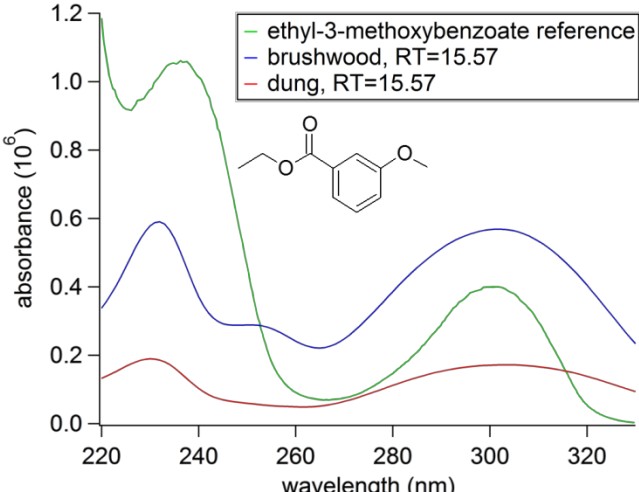

**Figure 10: UV-Vis absorption spectra from the PDA analysis of cookstove BBOA samples. The blue and red curves represent the background-subtracted absorbance at retention time of 15.57 min for brushwood-derived PM$_{2.5}$ and dung-derived PM$_{2.5}$, respectively. The reference absorption spectrum of ethyl-3-methoxybenzoate was reproduced from the NIST webbok database. The structure of ethyl-3-methoxybenzoate is pictured.**

**Table 1: List of common compounds found in all PM$_{2.5}$ samples regardless of fuel or stove type. Tentative molecular structure assignments are listed when the compound has previously been identified in the chemical biomass-burning literature, supported by the references in the last column. Count-based, normalized peak abundances are designated LOW (<1%), MEDIUM (1-9.99%), HIGH (100%). All species were detected as protonated ions.**

| Observed m/z | Calculated m/z | Chemical formula of neutral species | DBE | Relative average abundance | Tentative assignment(s) | References |
|---|---|---|---|---|---|---|
| 111.091 | 111.092 | C$_6$H$_{10}$N$_2$ | 3 | MEDIUM | | (Smith et al., 2009) |
| 121.064 | 121.065 | C$_8$H$_8$O | 5 | MEDIUM | | |
| 123.091 | 123.092 | C$_7$H$_{10}$N$_2$ | 4 | MEDIUM | | |
| 124.075 | 124.076 | C$_7$H$_9$ON | 4 | MEDIUM | | |
| 125.107 | 125.107 | C$_7$H$_{12}$N$_2$ | 3 | MEDIUM | | (Smith et al., 2009) |
| 133.075 | 133.076 | C$_8$H$_8$N$_2$ | 6 | MEDIUM | | (Laskin et al., 2009) |
| 134.071 | 134.071 | C$_7$H$_7$N$_3$ | 6 | MEDIUM | | (Laskin et al., 2009) |
| 137.059 | 137.060 | C$_8$H$_8$O$_2$ | 5 | HIGH | Anisaldehyde | (Simoneit et al., 1993; Smith et al., 2009) |
| 137.106 | 137.107 | C$_8$H$_{12}$N$_2$ | 4 | MEDIUM | | (Smith et al., 2009) |
| 138.090 | 138.091 | C$_8$H$_{11}$ON | 4 | LOW | | |
| 139.122 | 139.123 | C$_8$H$_{14}$N$_2$ | 3 | MEDIUM | | (Smith et al., 2009) |





| | | | | | | |
|---|---|---|---|---|---|---|
| 147.091 | 147.092 | $C_9H_{10}N_2$ | 6 | MEDIUM | | |
| 151.074 | 151.075 | $C_9H_{10}O_2$ | 5 | MEDIUM | Vinylguaiacol | |
| 151.122 | 151.123 | $C_9H_{14}N_2$ | 4 | MEDIUM | | |
| 153.138 | 153.139 | $C_9H_{16}N_2$ | 3 | HIGH | | |
| 159.091 | 159.092 | $C_{10}H_{10}N_2$ | 7 | MEDIUM | | (Laskin et al., 2009) |
| 160.075 | 160.076 | $C_{10}H_9ON$ | 7 | MEDIUM | | (Laskin et al., 2009) |
| 161.059 | 161.060 | $C_{10}H_8O_2$ | 7 | MEDIUM | | |
| 161.106 | 161.107 | $C_{10}H_{12}N_2$ | 6 | MEDIUM | | |
| 162.102 | 162.103 | $C_9H_{11}N_3$ | 6 | LOW | | |
| 163.074 | 163.075 | $C_{10}H_{10}O_2$ | 6 | MEDIUM | | |
| 165.138 | 165.139 | $C_{10}H_{16}N_2$ | 4 | MEDIUM | | |
| 167.069 | 167.070 | $C_9H_{10}O_3$ | 5 | HIGH | Veratraldehyde | (Simoneit et al., 1993) |
| 167.153 | 167.154 | $C_{10}H_{18}N_2$ | 3 | MEDIUM | | |
| 173.106 | 173.107 | $C_{11}H_{12}N_2$ | 7 | MEDIUM | | |
| 174.090 | 174.091 | $C_{11}H_{11}ON$ | 7 | MEDIUM | | |
| 175.074 | 175.075 | $C_{11}H_{10}O_2$ | 7 | MEDIUM | | |
| 175.122 | 175.123 | $C_{11}H_{14}N_2$ | 6 | MEDIUM | | |
| 177.053 | 177.055 | $C_{10}H_8O_3$ | 7 | MEDIUM | | |
| 177.090 | 177.091 | $C_{11}H_{12}O_2$ | 6 | MEDIUM | | |
| 177.101 | 177.102 | $C_{10}H_{12}ON_2$ | 6 | LOW | | |
| 177.137 | 177.139 | $C_{11}H_{16}N_2$ | 5 | LOW | | (Laskin et al., 2009) |
| 179.069 | 179.070 | $C_{10}H_{10}O_3$ | 6 | MEDIUM | Coniferaldehyde | |
| 179.153 | 179.154 | $C_{11}H_{18}N_2$ | 4 | MEDIUM | | |
| 181.169 | 181.170 | $C_{11}H_{20}N_2$ | 3 | HIGH | | |
| 183.090 | 183.092 | $C_{12}H_{10}N_2$ | 9 | HIGH | | |
| 183.184 | 183.186 | $C_{11}H_{22}N_2$ | 2 | MEDIUM | | |
| 186.090 | 186.091 | $C_{12}H_{11}ON$ | 8 | MEDIUM | | (Laskin et al., 2009) |
| 187.122 | 187.123 | $C_{12}H_{14}N_2$ | 7 | MEDIUM | | |
| 188.106 | 188.107 | $C_{12}H_{13}ON$ | 7 | MEDIUM | | |
| 189.101 | 189.102 | $C_{11}H_{12}ON_2$ | 7 | MEDIUM | | (Laskin et al., 2009) |
| 189.137 | 189.139 | $C_{12}H_{16}N_2$ | 6 | MEDIUM | | |
| 191.069 | 191.070 | $C_{11}H_{10}O_3$ | 7 | MEDIUM | | |
| 191.117 | 191.118 | $C_{11}H_{14}ON_2$ | 6 | LOW | | |
| 191.153 | 191.154 | $C_{12}H_{18}N_2$ | 5 | LOW | | |
| 193.085 | 193.086 | $C_{11}H_{12}O_3$ | 6 | MEDIUM | | |
| 193.169 | 193.170 | $C_{12}H_{20}N_2$ | 4 | MEDIUM | | |
| 197.106 | 197.107 | $C_{13}H_{12}N_2$ | 9 | MEDIUM | | |





| | | | | | | |
|---|---|---|---|---|---|---|
| 199.122 | 199.123 | $C_{13}H_{14}N_2$ | 8 | LOW | | |
| 200.106 | 200.107 | $C_{13}H_{13}ON$ | 8 | MEDIUM | | |
| 201.137 | 201.139 | $C_{13}H_{16}N_2$ | 7 | MEDIUM | | |
| 202.085 | 202.086 | $C_{12}H_{11}O_2N$ | 8 | MEDIUM | | (Laskin et al., 2009) |
| 203.117 | 203.118 | $C_{12}H_{14}ON_2$ | 7 | MEDIUM | | |
| 203.153 | 203.154 | $C_{13}H_{18}N_2$ | 6 | MEDIUM | | |
| 205.085 | 205.086 | $C_{12}H_{12}O_3$ | 7 | MEDIUM | | |
| 207.184 | 207.186 | $C_{13}H_{22}N_2$ | 4 | MEDIUM | | |
| 209.079 | 209.081 | $C_{11}H_{12}O_4$ | 6 | MEDIUM | | |
| 209.200 | 209.201 | $C_{13}H_{24}N_2$ | 3 | MEDIUM | | |
| 211.095 | 211.096 | $C_{11}H_{14}O_4$ | 5 | HIGH | Syringylethanone/ trimethoxyphenylethanone | (Simoneit et al., 1993) |
| 211.121 | 211.123 | $C_{14}H_{14}N_2$ | 9 | MEDIUM | | |
| 213.137 | 213.139 | $C_{14}H_{16}N_2$ | 8 | MEDIUM | | (Laskin et al., 2009) |
| 214.121 | 214.123 | $C_{14}H_{15}ON$ | 8 | MEDIUM | | |
| 215.153 | 215.154 | $C_{14}H_{18}N_2$ | 7 | MEDIUM | | |
| 216.100 | 216.102 | $C_{13}H_{13}O_2N$ | 8 | MEDIUM | | |
| 217.132 | 217.134 | $C_{13}H_{16}ON_2$ | 7 | MEDIUM | | |
| 217.168 | 217.170 | $C_{14}H_{20}N_2$ | 6 | MEDIUM | | |
| 219.100 | 219.102 | $C_{13}H_{14}O_3$ | 7 | MEDIUM | | |
| 227.153 | 227.154 | $C_{15}H_{18}N_2$ | 8 | MEDIUM | | |
| 229.132 | 229.134 | $C_{14}H_{16}ON_2$ | 8 | MEDIUM | | |
| 229.168 | 229.170 | $C_{15}H_{20}N_2$ | 7 | MEDIUM | | |
| 230.116 | 230.118 | $C_{14}H_{15}O_2N$ | 8 | MEDIUM | | |
| 231.147 | 231.149 | $C_{14}H_{18}ON_2$ | 7 | LOW | | |
| 232.095 | 232.097 | $C_{13}H_{13}O_3N$ | 8 | MEDIUM | | |
| 235.095 | 235.096 | $C_{13}H_{14}O_4$ | 7 | MEDIUM | | |
| 241.168 | 241.170 | $C_{16}H_{20}N_2$ | 8 | MEDIUM | | |
| 243.147 | 243.149 | $C_{15}H_{18}ON_2$ | 8 | MEDIUM | | |
| 243.184 | 243.186 | $C_{16}H_{22}N_2$ | 7 | LOW | | |
| 244.131 | 244.133 | $C_{15}H_{17}O_2N$ | 8 | MEDIUM | | |
| 246.111 | 246.112 | $C_{14}H_{15}O_3N$ | 8 | MEDIUM | | |
| 249.110 | 249.112 | $C_{14}H_{16}O_4$ | 7 | MEDIUM | | |

**Table 2: List of compounds found exclusively in the emissions from dung cookfires, regardless of stove type. The labels for the peak abundances are the same as in Table 1. All species unless otherwise noted were detected as protonated ions.**





| Observed $m/z$ | Calculated $m/z$ | Chemical formula of neutral species | DBE | Relative average abundance |
|---|---|---|---|---|
| 124.099 | 124.099 | $C_7H_{12}N_2{}^*$ | 3 | MEDIUM |
| 135.080 | 135.080 | $C_9H_{10}O$ | 5 | LOW |
| 135.092 | 135.092 | $C_8H_{10}N_2$ | 5 | MEDIUM |
| 136.076 | 136.076 | $C_8H_9ON$ | 5 | LOW |
| 137.071 | 137.071 | $C_7H_8ON_2$ | 5 | MEDIUM |
| 138.115 | 138.115 | $C_8H_{14}N_2{}^*$ | 3 | LOW |
| 141.138 | 141.139 | $C_8H_{16}N_2$ | 2 | LOW |
| 145.076 | 145.076 | $C_9H_8N_2$ | 7 | MEDIUM |
| 146.060 | 146.060 | $C_9H_7ON$ | 7 | MEDIUM |
| 146.084 | 146.084 | $C_9H_{10}N_2{}^*$ | 6 | LOW |
| 149.071 | 149.071 | $C_8H_8ON_2$ | 6 | LOW |
| 149.107 | 149.107 | $C_9H_{12}N_2$ | 5 | LOW |
| 151.086 | 151.087 | $C_8H_{10}ON_2$ | 5 | LOW |
| 152.107 | 152.107 | $C_9H_{13}ON$ | 4 | LOW |
| 152.130 | 152.131 | $C_9H_{16}N_2{}^*$ | 3 | LOW |
| 155.154 | 155.154 | $C_9H_{18}N_2$ | 2 | LOW |
| 160.099 | 160.099 | $C_{10}H_{12}N_2{}^*$ | 6 | LOW |
| 162.091 | 162.091 | $C_{10}H_{11}ON$ | 6 | LOW |
| 163.086 | 163.087 | $C_9H_{10}ON_2$ | 6 | MEDIUM |
| 163.123 | 163.123 | $C_{10}H_{14}N_2$ | 5 | MEDIUM |
| 164.107 | 164.107 | $C_{10}H_{13}ON$ | 5 | LOW |
| 169.076 | 169.076 | $C_{11}H_8N_2$ | 9 | HIGH |
| 169.170 | 169.170 | $C_{10}H_{20}N_2$ | 2 | MEDIUM |
| 171.091 | 171.092 | $C_{11}H_{10}N_2$ | 8 | MEDIUM |
| 172.075 | 172.076 | $C_{11}H_9ON$ | 8 | MEDIUM |
| 175.086 | 175.087 | $C_{10}H_{10}ON_2$ | 7 | MEDIUM |
| 176.070 | 176.071 | $C_{10}H_9O_2N$ | 7 | LOW |
| 176.107 | 176.107 | $C_{11}H_{13}ON$ | 6 | LOW |
| 176.118 | 176.118 | $C_{10}H_{13}N_3$ | 6 | LOW |
| 178.086 | 178.086 | $C_{10}H_{11}O_2N$ | 6 | LOW |
| 184.075 | 184.076 | $C_{12}H_9ON$ | 9 | MEDIUM |
| 185.107 | 185.107 | $C_{12}H_{12}N_2$ | 8 | MEDIUM |
| 187.086 | 187.087 | $C_{11}H_{10}ON_2$ | 8 | MEDIUM |
| 188.118 | 188.118 | $C_{11}H_{13}N_3$ | 7 | LOW |



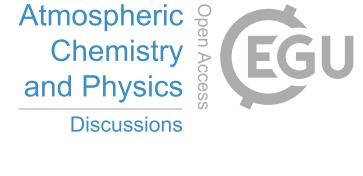

| | | | | |
|---|---|---|---|---|
| 189.091 | 189.091 | $C_{12}H_{12}O_2$ | 7 | LOW |
| 190.086 | 190.086 | $C_{11}H_{11}O_2N$ | 7 | MEDIUM |
| 190.133 | 190.134 | $C_{11}H_{15}N_3$ | 6 | MEDIUM |
| 191.081 | 191.082 | $C_{10}H_{10}O_2N_2$ | 7 | LOW |
| 192.102 | 192.102 | $C_{11}H_{13}O_2N$ | 6 | LOW |
| 193.133 | 193.134 | $C_{11}H_{16}ON_2$ | 5 | LOW |
| 195.091 | 195.092 | $C_{13}H_{10}N_2$ | 10 | MEDIUM |
| 195.185 | 195.186 | $C_{12}H_{22}N_2$ | 3 | HIGH |
| 197.201 | 197.201 | $C_{12}H_{24}N_2$ | 2 | MEDIUM |
| 198.091 | 198.091 | $C_{13}H_{11}ON$ | 9 | MEDIUM |
| 198.102 | 198.103 | $C_{12}H_{11}N_3$ | 9 | LOW |
| 199.086 | 199.087 | $C_{12}H_{10}ON_2$ | 9 | MEDIUM |
| 200.118 | 200.118 | $C_{12}H_{13}N_3$ | 8 | LOW |
| 201.102 | 201.102 | $C_{12}H_{12}ON_2$ | 8 | MEDIUM |
| 202.122 | 202.123 | $C_{13}H_{15}ON$ | 7 | LOW |
| 202.133 | 202.134 | $C_{12}H_{15}N_3$ | 7 | LOW |
| 204.101 | 204.102 | $C_{12}H_{13}O_2N$ | 7 | LOW |
| 204.149 | 204.150 | $C_{12}H_{17}N_3$ | 6 | MEDIUM |
| 205.097 | 205.097 | $C_{11}H_{12}O_2N_2$ | 7 | LOW |
| 205.133 | 205.134 | $C_{12}H_{16}ON_2$ | 6 | MEDIUM |
| 205.169 | 205.170 | $C_{13}H_{20}N_2$ | 5 | MEDIUM |
| 206.117 | 206.118 | $C_{12}H_{15}O_2N$ | 6 | LOW |
| 207.112 | 207.113 | $C_{11}H_{14}O_2N_2$ | 6 | LOW |
| 207.149 | 207.149 | $C_{12}H_{18}ON_2$ | 5 | LOW |
| 209.107 | 209.107 | $C_{14}H_{12}N_2$ | 10 | MEDIUM |
| 209.128 | 209.128 | $C_{11}H_{16}O_2N_2$ | 5 | LOW |
| 211.144 | 211.144 | $C_{11}H_{18}O_2N_2$ | 4 | MEDIUM |
| 212.106 | 212.107 | $C_{14}H_{13}ON$ | 9 | MEDIUM |
| 212.118 | 212.118 | $C_{13}H_{13}N_3$ | 9 | LOW |
| 214.086 | 214.086 | $C_{13}H_{11}O_2N$ | 9 | MEDIUM |
| 215.117 | 215.118 | $C_{13}H_{14}ON_2$ | 8 | MEDIUM |
| 216.149 | 216.150 | $C_{13}H_{17}N_3$ | 7 | LOW |
| 217.085 | 217.086 | $C_{13}H_{12}O_3$ | 8 | LOW |
| 217.097 | 217.097 | $C_{12}H_{12}O_2N_2$ | 8 | MEDIUM |
| 218.103 | 218.104 | $C_{10}H_{11}ON_5$ | 8 | LOW |
| 218.117 | 218.118 | $C_{13}H_{15}O_2N$ | 7 | LOW |
| 218.165 | 218.165 | $C_{13}H_{19}N_3$ | 6 | LOW |



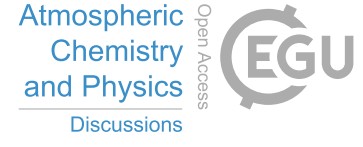

| | | | | |
|---|---|---|---|---|
| 219.112 | 219.113 | $C_{12}H_{14}O_2N_2$ | 7 | MEDIUM |
| 219.149 | 219.149 | $C_{13}H_{18}ON_2$ | 6 | LOW |
| 219.185 | 219.186 | $C_{14}H_{22}N_2$ | 5 | MEDIUM |
| 221.080 | 221.081 | $C_{12}H_{12}O_4$ | 7 | LOW |
| 221.128 | 221.128 | $C_{12}H_{16}O_2N_2$ | 6 | MEDIUM |
| 221.201 | 221.201 | $C_{14}H_{24}N_2$ | 4 | MEDIUM |
| 223.122 | 223.123 | $C_{15}H_{14}N_2$ | 10 | MEDIUM |
| 223.216 | 223.217 | $C_{14}H_{26}N_2$ | 3 | MEDIUM |
| 224.107 | 224.107 | $C_{15}H_{13}ON$ | 10 | LOW |
| 225.102 | 225.102 | $C_{14}H_{12}ON_2$ | 10 | MEDIUM |
| 225.138 | 225.139 | $C_{15}H_{16}N_2$ | 9 | MEDIUM |
| 226.122 | 226.123 | $C_{15}H_{15}ON$ | 9 | MEDIUM |
| 227.117 | 227.118 | $C_{14}H_{14}ON_2$ | 9 | MEDIUM |
| 228.101 | 228.102 | $C_{14}H_{13}O_2N$ | 9 | MEDIUM |
| 228.138 | 228.138 | $C_{15}H_{17}ON$ | 8 | MEDIUM |
| 230.164 | 230.165 | $C_{14}H_{19}N_3$ | 7 | LOW |
| 231.112 | 231.113 | $C_{13}H_{14}O_2N_2$ | 8 | MEDIUM |
| 232.133 | 232.133 | $C_{14}H_{17}O_2N$ | 7 | LOW |
| 233.128 | 233.128 | $C_{13}H_{16}O_2N_2$ | 7 | LOW |
| 233.164 | 233.165 | $C_{14}H_{20}ON_2$ | 6 | LOW |
| 233.201 | 233.201 | $C_{15}H_{24}N_2$ | 5 | MEDIUM |
| 235.216 | 235.217 | $C_{15}H_{26}N_2$ | 4 | MEDIUM |
| 237.138 | 237.139 | $C_{16}H_{16}N_2$ | 10 | MEDIUM |
| 239.117 | 239.118 | $C_{15}H_{14}ON_2$ | 10 | MEDIUM |
| 239.153 | 239.154 | $C_{16}H_{18}N_2$ | 9 | MEDIUM |
| 241.133 | 241.134 | $C_{15}H_{16}ON_2$ | 9 | MEDIUM |
| 242.117 | 242.118 | $C_{15}H_{15}O_2N$ | 9 | LOW |
| 243.112 | 243.113 | $C_{14}H_{14}O_2N_2$ | 9 | LOW |
| 244.096 | 244.097 | $C_{14}H_{13}O_3N$ | 9 | LOW |
| 245.128 | 245.128 | $C_{14}H_{16}O_2N_2$ | 8 | MEDIUM |
| 245.164 | 245.165 | $C_{15}H_{20}ON_2$ | 7 | MEDIUM |
| 247.143 | 247.144 | $C_{14}H_{18}O_2N_2$ | 7 | LOW |
| 247.216 | 247.217 | $C_{16}H_{26}N_2$ | 5 | MEDIUM |
| 249.232 | 249.233 | $C_{16}H_{28}N_2$ | 4 | MEDIUM |
| 251.153 | 251.154 | $C_{17}H_{18}N_2$ | 10 | LOW |
| 253.133 | 253.134 | $C_{16}H_{16}ON_2$ | 10 | LOW |
| 255.112 | 255.113 | $C_{15}H_{14}O_2N_2$ | 10 | LOW |



| 255.148 | 255.149 | C₁₆H₁₈ON₂ | 9 | LOW |
|---|---|---|---|---|
| 255.185 | 255.186 | $C_{17}H_{22}N_2$ | 8 | LOW |
| 258.112 | 258.112 | $C_{15}H_{15}O_3N$ | 9 | LOW |
| 259.143 | 259.144 | $C_{15}H_{18}O_2N_2$ | 8 | LOW |
| 259.180 | 259.180 | $C_{16}H_{22}ON_2$ | 7 | LOW |
| 269.127 | 269.128 | $C_{16}H_{16}O_2N_2$ | 10 | LOW |
| 283.143 | 283.144 | $C_{17}H_{18}O_2N_2$ | 10 | LOW |

*species detected as an ion-radical

**Table 3: The list of rentention times, absorption peak maxima, and chemical formulas of the BrC chromophores detected in the brushwood smoke sample. Tentative assignments are given based on compounds previously identified in the lignin pyrolysis literature.**

| LC retention time (min) | $\lambda_{max}$ (nm) | Nominal molecular weight (amu) | Chemical formula(s) | Tentative assignment |
|---|---|---|---|---|
| 6.26 | 383 | 192 | $C_9H_8N_2O_3$ | |
| 7.15 | 392 | 141 | $C_7H_8O_3$ | |
| 10.55 | 305 | 183 | $C_9H_{10}O_4$ | Homovanillic acid/syringealdehyde |
| 13.29 | 265 | 155 | $C_8H_{10}O_3$ | Syringol |
| 14.44 | 305 | 169 | $C_8H_8O_4$ | Vanillic acid |
| | | 183 | $C_9H_{10}O_4$ | Homovanillic acid/syringealdehyde |
| 15.57 | 299 | 181 | $C_{10}H_{12}O_3$ | Ethyl-3-methoxybenzoate |
| | | 167 | $C_9H_{10}O_3$ | Veratraldehyde |
| 16.95 | 313, 334 | 186 | $C_{11}H_7NO_2$ | |
| 17.25 | 331 | 165 | $C_9H_8O_3$ | |
| | | 162 | $C_9H_7NO_2$ | |
| 18.13 | 341 | 209 | $C_{11}H_{12}O_4$ | |
| 18.32 | 229, 337 | 179 | $C_{10}H_{10}O_3$ | |
| 19.78 | 305, 330 | 194 | $C_{10}H_{10}O_4$ | Ferulic acid |
| 24.11 | 290, 330 | 259 | $C_{15}H_{14}O_4$ | |
| 28.07 | 334 | 184 | $C_8H_9NO_4$ | |
| 29.24 | 330 | 198 | $C_{13}H_{10}O_2$ | |
| | | 230 | $C_{13}H_{10}O_4$ | |
| 33.81 | 340 | 227 | $C_{14}H_{10}O_3$ | |





**Table 4: The list of retention times, absorption peak maxima, and chemical formulas of the BrC chromophores detected in the the dung smoke sample. Tentative assignments are given based on compounds previously identified in the lignin pyrolysis literature.**

| LC retention time (min) | $\lambda_{max}$ (nm) | Nominal molecular weight (amu) | Chemical formula(s) | Tentative assignment |
|---|---|---|---|---|
| 8.50 | 295 | 167 | $C_8H_9NO_3$ | |
| 9.09 | 282,300 | 166 | $C_9H_{10}O_3$ | |
| | | 168 | $C_8H_8O_4$ | |
| 10.59 | 252, 289, 393 | 182 | $C_9H_{10}O_4$ | Homovanillic acid/syringealdehyde |
| 12.22 | 282 | 122 | $C_7H_6O_2$ | Benzoic acid |
| 14.44 | 306 | 168 | $C_8H_8O_4$ | Vanillic acid |
| | | 182 | $C_9H_{10}O_4$ | Homovanillic acid/syringealdehyde |
| | | 164 | $C_9H_8O_3$ | |
| 15.57 | 300 | 174 | $C_{10}H_{12}O_3$ | Ethyl-3-methoxybenzoate |
| | | 166 | $C_9H_{10}O_3$ | Veratraldehyde |
| 16.35 | 286 | 174 | $C_{11}H_{10}O_2$ | |
| 18.28 | 290, 330[a] | 162 | $C_{10}H_{10}O_2$ | |
| 19.5 | 323[a] | 220 | $C_{12}H_{12}O_4$ | |
| 19.72 | 331[a] | 194 | $C_{10}H_{10}O_4$ | Ferulic acid |
| 20.85 | 352[a] | 188 | $C_{12}H_{12}O_2$ | |
| 24.54 | 299, 308 | 178 | $C_{10}H_{10}O_3$ | |
| 25.28 | 290, 320 | 218 | $C_{12}H_{10}O_4$ | |
| 29.17 | 332 | 198 | $C_{13}H_{10}O_2$ | |
| | | 230 | $C_{13}H_{10}O_4$ | |
| 29.60 | 358[a] | 213 | $C_{13}H_9O_3$ | |

[a] signifies a shoulder, rather than a clear peak