# Peer review of "Molecular Composition of Particulate Matter Emissions from Dung and Brushwood Burning Household Cookstoves in Haryana, India"

_Atmospheric Chemistry and Physics, 2017_

## Referee Comment (RC1) · R. J. Yokelson (Referee) · 1 Oct 2017

Review of Molecular Composition of Particulate Matter Emissions from Dung and Brushwood Burning Household Cookstoves in Haryana, India, by Fleming et al.

By Bob Yokelson

The authors have successfully identified numerous individual chemical species that absorb UV light in an important, but under-studied type of biomass burning aerosol (wood and dung cooking fires). The authentic cooking fire samples are difficult to obtain and the author's samples were subsequently analyzed off-line with a unique, extremely powerful array of coupled optical and mass-spectral techniques. The team has a great deal of relevant expertise in all aspects of the study and it includes leading practitioners of these advanced analysis techniques. The large body of data represents a significant investment of effort and the work should definitely be published. With some straightforward improvements the paper could be very good. I provide an overview followed by specific suggestions referenced to page and line number in order of appearance.

Overview:

More information is needed about the sampling, calculations, and error budget in the paper. A few additional sampling details and equations are given in the SI, but more are needed and should be in the main paper. Currently, the emission factors are said to be "orders of magnitude" upper limits, but it's proposed via limited comparisons that the MAC are "OK" – though they are given without an error estimate. The final MAC seem potentially "high" to me and their application should be clarified. Along with more detail on the sampling, the associated uncertainty in each step should be discussed near the beginning of the paper so the context and limitations are clear as the discussion is read, rather than reader wondering and only potentially finding out later in the paper.

I suggest a clear and commonly-used format would be to clarify how each step is done and discuss/estimate individually the different sources of error of each of these steps in order. For instance:

1. What were the inlet positions w.r.t. to the fires? Flaming produces a greater vertical velocity and flux of products than smoldering. Gas-particle partitioning is proportional to particle concentration and is also temperature dependent. Further from the source the smoke has diluted and cooled to some degree and vertical velocity doesn't need to be measured. A detailed diagram with distances, temperatures, concentrations (description of any dilution?), etc should be given in the paper.

2. What were the relative positions of the canister and filter inlets? Were they close or in well-mixed smoke?

3. What was the relative timing of the canister and filter sampling? Canisters tend to fill quickly at a non-constant rate while the filters were acquired over the whole fire.

4. When the results from two filters are coupled, what was the spatial and temporal overlap of the filter collection?

5. The authors evidently measured extraction efficiency once at 50%, but did not specify 50% of what (PM2.5, OA, etc)? Nor is it stated if the extraction efficiency is the same for all chromophores. Later in paper it's stated that the extraction efficiency was "lower sometimes." How much lower? Maybe give a range and consult some studies where extraction efficiency has been estimated by comparison to some familiar term e.g. OA measured by AMS or conventional OC analyses.

6. Ionization efficiency is maybe stated near the end of the paper to be higher for polar compounds, but some polar compounds that are ubiquitous in BBOA like levoglucosan were not seen in some samples. Can this be explained?

7. Given the realistic limitations of any technique, in the introduction or after the fleshed-out experimental/error section then maybe a brief summary of the strengths and weaknesses of various approaches to understanding aerosol optical properties is in order to provide context for readers. For instance, my understanding is that laser-based techniques would be well-suited for measuring the overall absorption of real aerosol (after drying) that contains BC, BrC, and other species at specific wavelengths, but non-power law features are an issue in fitting the cross-section at unmeasured wavelengths, they certainly cannot identify individual compounds, and they have some uncertainty in differentiating between BrC and coating effects (e.g. Pokhrel et al., 2017). Extractions (I think) eliminate BC and coating effects so that only BrC is probed. Following with off-line analysis by broadband UV absorption, retention times, and exact mass is very powerful for measuring the true BrC spectral shape and compound identification. But quantification and how the MAC relate to real-world aerosol is not clear from the paper now.

8. In light of above, provide some at least brief, rough guidance on how the overall optical properties of real BC-containing aerosol (coating effects and all) could be estimated from the extraction results that are presented. It seems possible using independently measured BC emissions, but I did not find that in the paper.

9. Make sure to specify what is being measured throughout (e.g. the MACs are absorption of what per mass of what?) and provide uncertainties.

10. BBOA almost certainly contains 1000's of species. The mass spectra obtained depend on variable detection limits and loading and natural variability makes the number of samples important. Thus, throughout the paper, I would refer to "observed" complexity, with actual complexity beyond the capabilities of current instruments.

11. Some new species are observed, but many species commonly observed by other techniques are missing some or all of the time. It would be helpful to clarify which off-line compound identification techniques access which types of chemical space well and, if possible, the relative mass contributions to total OA measured by various techniques. This could be a sentence or two in the paragraph recommended in point 7 above.

12. Tracers and markers are not the same thing and these terms are frequently misused. Tracers are emitted in a narrow, well-characterized ratio to the observable of interest and can be used to quantify impacts at receptor sites. Markers are useful but typically emitted in highly variable amounts by unique sources and useful qualitatively.

Comments on text in order of appearance:

P1, L13: "organic particles" should be "organic matter in particles" or "organic aerosol" to allow for the possibility of internally mixed particles.

P1, L20: "selected "extractable"" or similar seems appropriate before "compounds" or "numerous" instead of "selected"

P1, L21: The fact that many of these species are newly observed will make the reader curious if they are newly observed because they are relatively rare or because most of the organic matter in BBOA was previously un-speciated. If that question can be answered it would be great to do so in the paper somewhere. A related fine point is that if most of the observed compounds are newly seen despite the fact that numerous other compounds have already been seen in BBOA, then it is unlikely that this study probes the true molecular complexity. This is an empirical, observed molecular complexity impacted by detection limits and loading.

P1, L22: "stove-specific combustion conditions" might be better than "stove" if the stoves impact emissions by impacting the mix combustion processes?

P1, L23: "emission factor" and "observed molecular complexity"

P1, L25-7: These MACs seem ~2x too high compared to other studies if they are referenced to g of PM2.5, especially as a lower limit? And it should be clear what they are for; real aerosol with BC included or just the BrC? If BC is added back to represent real aerosols, how is that done?

P2, L12: "depends to some extent" since amount of PM and individual susceptibility to various toxins matter a lot.

P2, L20: lifetime of BrC also important.

P2, L24: probably don't need same reference twice?

P2, L25: "a photoacoustic spectrometer" should be "photoacoustic extinctiometers (PAX)" (throughout) and "895" should be "870", L26: "cook" > "cooking"

P2, L26-29: The EFs for BrC quoted from Stockwell et al are actually their EF Babs at 405 nm (not the same thing as explained next). These include absorption by BC and BrC. Different EFabs, tentatively for just the BrC are also provided though.

Some relevant background on Stockwell et al 2016 from the corresponding author:

Lack and Langridge (2013, Table 1) recommended an MAC for "BrC," but actually meant an MAC referenced to the mass of "BrC-containing OA." They clarified that their MAC was an average value for BBOA but that the MAC can range a lot. The OA MAC was later found to depend on BC/OA by Saleh et al 2014. We used the concept of an EF for "BrC" based on the Lack and Langridge MAC in Stockwell et al, but have since abandoned that terminology as we think it is too easy to misinterpret. Meanwhile, in the Stockwell paper cited here we tried to present qualifying text as follows: "The BrC mass calculated this way is considered roughly equivalent to the total organic aerosol (OA) mass, which as a whole weakly absorbs UV light, and not the mass of the actual chromophores. The MAC of bulk OA varies substantially and the BrC mass we calculate with the single average MAC that we used is only qualitatively similar to bulk OA mass for "average" aerosol and even less similar to bulk OA for non-average aerosol (Saleh et al., 2014). The BrC mass estimated by PAX in this way was independently sampled and worth reporting, but the filters and mAMS provide additional samples of the mass of organic aerosol emissions that have lower per-sample uncertainty for mass. Most importantly, the optical properties from the PAX (SSA, AAE, and absorption EFs calculated as detailed below) are not impacted by MAC variability or filter artifacts."

Thus, our EF BC and the values listed just above are the best to compare to in Stockwell et al. Thanks to the authors, we rechecked our cooking fire table, found one error that only impacts the SSAs, and have posted a corrigendum. (We discovered that the SSA labels in Table 4, but not Table S8 had been reversed: it should be SSA 870 and then SSA 405 below that in Tab 4.)

Then finally, the more robust estimates of speciated PM2.5 mass from the same study that we alluded to in Stockwell et al., 2016 can now be found in Jayarathne et al. (2017) and are useful for comparisons with this work as will be pointed out.

Jayarathne, T., Stockwell, C. E., Bhave, P. V., Praveen, P. S., Rathnayake, C. M., Islam, Md. R., Panday, A. K., Adhikari, S., Maharjan, R., Goetz, J. D., DeCarlo, P. F., Saikawa, E., Yokelson, R. J., and Stone, E. A.: Nepal Ambient Monitoring and Source Testing Experiment (NAMaSTE): Emissions of particulate matter from wood and dung cooking fires, garbage and crop residue burning, brick kilns, and other sources, Atmos. Chem. Phys. Discuss., https://doi.org/10.5194/acp-2017-510, in review, 2017.

P2, L31-32: Just an observation that the dung is expected to have lower MAC than wood consistent with lower BC/OA per Saleh et al 2014.

P3, L13: Should "The" which could imply "all" be "Many"?

L17-19: This second half of the sentence starting with "while" is unclear in and also ceanothus is from US NW.

P3, L21: N accounts for a small mass fraction of BB PM2.5 so N-containing organics are likely a small fraction of the total BBOA?

P3, L25: eliminate "The most" since that will quickly be dated?

P3, L32: "more" or "additional"?

P4, L4-5: I'm sure this is worthwhile data regardless if it is the first or last data, but it seems unlikely this would be the first detailed study of brushwood or dung smoke since these sources have been studied intensively for > twenty years. A recent example that contains examples of historical references is Jayarathne et al. (2017). Simoneit et al have been characterizing smoke for > 20 years.

P4, L23-30: The pictures are useful, but Fig 1A currently shows a concoction of unidentified tubing and a diagram is also needed. Questions arise with some also in the overview. How did the authors ensure representative sampling of well-mixed emissions for all devices? For instance, was the data corrected for the different vertical velocity in the smoke column above the fire? This is important because during flaming, the flux of emissions can be much greater than during smoldering. What were the flow rates and residence times in inlets, was any dilution used, were the downstream filters side by side or in series, were the pumps downstream of the filters, how were filters stored during the 25-30 hours not at -80 C (in a cooler with dry or blue ice or at ambient T, what was ambient T if relevant?), were backgrounds or field blanks taken, error in gravimetric analysis, etc?

P4, L24: "BBOA" > "PM2.5"

P5, L1-9: The next step after filter collection and before mass spec is filter extraction, i.e. how was extraction done, what is extraction efficiency compared to total BBOA and is it the same for all the species detected, etc? As written it sounds as if the filter is inside a capillary. It may not be possible to estimate extraction efficiency if done by a droplet flowing over the filter surface? But the bottom line should be clear.

P5, L6-7: Why positive mode and are there species only seen in negative ion mode?

P5, L7 What are MRFA and Ultramark, what is the relevance of the calibration species, how were the cal results applied?

P5, L8-10: Filter deposits may not be uniform. What percent of peaks were seen in only one sample or had $S:N < 3$. This is useful context that helps relate reported complexity to observed complexity, which is itself a subset of actual complexity.

P5, L12: Does Kendrick analysis with CH2 and "H2" base units help with O-containing species?

P5, L13: "mass-calibrated"

P5, L16: About what percent of signal or number of peaks was above m/z 350?

P5, L20: K is normally the most abundant alkali metal in BB-PM (not Na). S and Cl can be high in dung (Hosseini et al and numerous other papers). Brief explanation of why not included?

P5, L22: No "O" in the formula, but it is in the explanation?

P5, L27: Why is the extraction solvent mixture different here, i.e. not containing water? What fraction of mass is extracted and of what types of compounds (in summary form)?

P6, L16: It should be specified that this is an MAC for the extracted BrC only. Are the units (cm) consistent with the reported MAC (m)? "Cmass" is the "solution mass concentration" of what and how measured?

P6, L17-18: Another filter collected where and when? Here is an example why the spatial and temporal overlap of the filter collection is important to describe in experimental section. How are PM2.5 on the other filter and "Cmass" connected?

P6, L20: So is the target mass reference for the MAC PM2.5 mass then?

P6, L20-21: How do they do know the mass extraction efficiency was < 50% sometimes and why would a lower mass extraction mean the MACs are a lower limit? What if the extraction got all the chromophores, but only half the total mass, then would the raw MAC be a factor of two high? It seems the impact on the MAC would depend on the relative extraction efficiency of the chromophores and other constituents.

P6, L21: Again it seems the AAE are for the BrC component only? Can the authors add a sentence on how their AAE and MAC can be adjusted to represent real aerosol that also contains BC?

P6, L27: Abundance is usually used to indicate how many there are of an item. Peak area or height is usually used to estimate the signal strength or amount of compound. Should "abundance" be replaced by "area" or something else?

P6, L27-29: So if I understand this and the SI right, the EFPM(total) was estimated separately (in a sparsely described experiment) and then EFPM(total) was partitioned among the peaks observed on the MS according their relative intensity. If so this should be stated as a sentence in the main text. It's a concern that the real EF could be ten times or even several "orders" of magnitude lower. Should the authors reconsider even reporting EF directly as such? Maybe it's safer to label them as "upper limit EF in the tables and figures to spare potential future misinterpretation?

Also, can the error be reduced or the error budget be tightened up? For instance, the authors may detect some species that have been better quantified from these sources in other studies? Can they use their ratios to any overlap compound with the extensive quantitative analyses reported in Jayarathne et al 2017 and many others? Ratios to levoglucosan or PAHs for instance may be helpful to constrain the "EFs" to a realistic range?

P6, L28-30 and Figure 2 comments: It's true that authors show non-identical spectra from the sources even though the two dung spectra look pretty similar. But proof of distinct signatures requires enough samples to quantify the variability in each source. Right now it's not clear if the method uncertainty is larger or smaller than natural variability or what the observed variability is. The authors should try to characterize observed source to source variation with some metric (# or % of unique peaks) and this is done to some extent later in the paper, which is good. As noted above, the y-axis label should be "Upper limit EF" or "approx. relative abundance" Relevant to earlier comments, the spectra are too simple to represent all the components of BBOA although more details might be revealed with a log scale.

P7, L8: Dung has much higher N-content than wood and that makes excellent sense based on known plant and animal physiology. The Gautam reference has much higher N than normal for wood. See Stockwell et al 2016 for dung and Coggon et al Fig 3 for wood or many other sources referenced in these papers.

P7, L8-11: This sentence doesn't quite make sense. Typo? Punctuation?

P7, L12: Could provide a few key citations on lignin pyrolysis – there are many. It would be interesting if the authors could show that the cellulose preferentially ends up in the gas phase? But maybe the method has low sensitivity for sugars (from cellulose), which are usually abundant in BBOA (Christian et al., 2010; Jayarathne et al., 2017)?

P7, L 14: Dung and embedded grasses are both high in Cl content (Stockwell et al., 2016 and references there-in, especially Lobert et al., 1999) so expect high Cl from cooking with dung and ag residues as in Stockwell et al. 2014-2015 ACP papers.

P7, L15-16: Clarify if these large peaks were included when the EF was partitioned?

P7, L21-26: K is well-known to be enhanced in biomass (Table 1 in Hosseini et al., 2013) and K has a very long history as a biomass burning "tracer" (e.g. Sullivan et al., 2014 and references there-in). It is well-known that K is primarily emitted by flaming while levoglucosan is primarily emitted by smoldering. If the stoves had different flaming/smoldering ratios that could explain variability in K production.

Hosseini, S., Urbanski, S., Dixit, P., Li, Q., Burling, I., Yokelson, R., Johnson, T., Shrivastava, M. K., Jung, H., Weise, D., Miller, W., and Cocker III, D.: Laboratory characterization of PM

emissions from combustion of wildland biomass fuels, J. Geophys. Res., 118, 9914–9929, doi:10.1002/jgrd.50481, 2013.

Sullivan, A. P., May, A. A., Lee, T., McMeeking, G. R., Kreidenweis, S. M., Akagi, S. K., Yokelson, R. J., Urbanski, S. P., and Collett Jr., J. L.: Airborne characterization of smoke marker ratios from prescribed burning, Atmos. Chem. Phys., 14, 10535-10545, doi:10.5194/acp-14-10535-2014, 2014.

P7, L27-30: There is a difference between a tracer and a marker and levoglucosan (LG, a cellulose pyrolysis product) is the latter. If LG was a good tracer then it would be in all the samples and at reproducible amounts. LG is in fact normally found to be a major, but variable component of BBOA (Sullivan et al 2014; Jayarathne et al 2017; Christian et al 2010). If LG is missing from many of the samples that suggests detection issues that could cause some of the BBOA species to escape the analysis procedures used.

P7, L31 - P8, L2: There were more than three spectra of each stove/fuel combo, but three were chosen how? Then peaks not appearing in all three spectra of a combo were discarded why? Then the remaining peaks were scaled and the spectra for each stove/fuel combo were averaged together. I think that is the right order, which may be jumbled in the text?

P8, L6-12: First compounds found from all three cooking types are listed. Then compounds only found from brushwood are discussed. That I can follow. Then the compounds that are found in all dung cooking. But not unique to dung cooking? It's not clear here what the difference between sections 3.5.1 and 3.5.2 is. I.e. what is the difference between "common to" and "detected in all"?

P8, L11-12: If the stove material caused the emissions it potentially might not matter what the fuel is since all fires heat the stove.

P8, L14-17: The true number of constituents for all the PM types is much greater than observed so maybe just say ~we saw more peaks from A than B.

P8, L15-30: Can this overview of the results section be re-phrased or re-organized to make it easier to follow?

P9, L4-5: "detected elemental" and the relative insensitivity for sugars, or anything else needs to be discussed earlier – maybe in the introduction or at latest the experimental section.

P9, L6-8: I don't believe the %N in Gautam et al, it goes against all the other studies I've seen dating back to Susott et al., 1996, unless Gautam et al included the foliage with their wood.

P9, L17: All biomass is ~25% lignin, 25% hemicellulose, and ~50% cellulose polymers though the monomer units differ.

P9, L18: delete "at"

P9, L18-19: by "commonly detected" do they mean by their group or are there references to other groups? There are numerous studies that characterized BBOA.

P9, L20-21: "found" to "observed" better?

P9, L25: "coniferyl" alcohol and that comes only from conifers whereas the others are unique to hardwoods or grasses, which are probably more relevant in India.

P9, L30: In the experimental overview, the range of ionization efficiencies could be provided?

P9, L31: Jayarathne et al also discuss species unique to dung

P10, L1: important to qualify "the observed chemical …was far more complex" etc

P10, L2: By "reproducibly" it means "seen" in all "n" samples, but not in the same ratio to total PM2.5? That should be clear if so.

P10, L5-6: This is a little hard to follow as the authors use "found in all dung …" in the section header, then next mention "detected exclusively" in one type or another, and then "combine all…" Maybe change "Hereafter" to "However" would help with the transition?

P10, L10 - P11, L6; Figures 5 -7: Jayarathne et al 2017 and references there-in quantified numerous PAHs for similar fuels, which may be helpful to compare to.

P11, L8: At the outset useful to state what these MACs represent: absorption due to extractables only per PM2.5 mass? I think the mass reference is not extract mass or mass of chromophores. It's important in the larger context if they are MAC that don't include the BC component or any un-extracted chromophores.

P11, L8-11: "browner" OA from wood cooking makes sense empirically given the higher EC or BC to OA ratio for these fires per Saleh et al., 2014, Jayarathne et al., 2017. The latter reference also reports higher PM emissions from dung cooking than wood cooking in agreement with cited previous work. Their measured and cited EFPM2.5 from South Asia for wood and dung cooking may be useful to compare to.

P11, L12: The MAC is the absorption coefficient per mass of PM. In order to convey the absorption per unit fuel consumption, "coefficient by" should probably be replaced by "emission factor of"

P11, L 11-13: On the overall absorption per unit fuel consumption. This was evaluated by Stockwell et al., (2016) for particles containing BC and BrC and, with higher uncertainty, for just the BrC component, with the following results (in $m^2$/kg) (variability also shown in reference).

|  | Wood | Dung |
| --- | --- | --- |
| EF Babs 405 | 10.6 | 5.85 |

EF Babs-405-BrC       8.40            5.43

EF Babs 870            1.04            .197

Wood-burning aerosol absorbed about 5 x more per kg burned at 870 because of the higher BC emissions, but just 2 x more at 405 for the overall particles. More BrC absorption per unit fuel consumption was observed on average for wood, but within variability the amounts overlapped. Can the authors use BC/EC EFs for these fire types to get their own estimates for absorption EFs for real fires?

P11, L14-21: Throughout this section, for this work and other work, the mass reference (PM2.5, OA, etc?) should be rechecked and specified and uncertainties for the MACs should be provided.

It makes sense that the brushwood MAC is larger than the dung MAC given the BC/OA dependence of MAC described by Saleh et al. (2014) as noted above.

At a cursory glance, it seems like the MAC values at ~400 nm that the authors selected for comparisons are on average higher than values in Lack and Langridge, the extensive tables in Olson et al 2015, or Bluvshtein et al., (2017). It could be helpful if the authors could include these studies in their discussion and comment on any implications there may be.

Olson, M. R., Garcia, M. V., Robinson, M. A., Van Rooy, P., Dietenberger, M. A., Bergin, M., and Schauer, J. J.: Investigation of black and brown carbon multiple-wavelength-dependent light absorption from biomass and fossil fuel combustion source emissions, J. Geophys. Res.: Atmos., 120, 6682-6697, doi:10.1002/2014JD022970, 2015.

P11, L22-26: If my understanding is correct, one advantage of extraction techniques is that the BC is eliminated along with uncertainties in BrC attribution due to BC coatings or AAE. If that is right, the authors should clarify this is an AAE for the BrC only. Also, the AAEs can depend on the choice of wavelengths fit with a power law and this study has much better wavelength coverage than just the 2-3 wavelengths commonly used in optical in-situ approaches. The overall AAE with BC included is also important to describe real in-situ aerosol. The AAE with BC included may be lower since the AAE of BC is near 1. Stockwell et al obtained AAEs of 3 and 4.6 for wood and dung, respectively using an optical in-situ approach on aerosol containing BC. Can the authors estimate overall absorption values for real intact BC-containing aerosol from their extract values? It seems doable using EFBC and BC optical properties.

P11, L25: Why is BC mentioned? Per above, isn't it eliminated in the extraction process? It's not clear why the authors say the following: "However, the observed absorption can be definitively attributed to BrC since AAE values of 2 or greater indicate that light absorption comes from BrC as opposed to BC (Kirchstetter et al., 2004; Laskin et al., 2015)." Pure uncoated BC has an AAE near 1, but any aerosol AAE can have some contribution from BC if BC is present. I think the

definitive attribution comes from the fact that BC and BC coating effects are eliminated in the extraction.

P11, L27: Move first sentence to P12, L17?

P12, L4: "lignin-derived"?

P12: L3-16: This is great stuff. I wonder if retention times were measured for standards to support identifications.

P12: L22-28: On line 24, why were the early eluters ignored? Combining line 22 and line 26, is the conclusion that polar compounds tend to be detected more efficiently, but elute earlier and may be ignored?  Minor tweaks to text could likely clarify this section.

P12, L29 – P13, L2: It's neat that absorption features can be used with mass and retention times to support compound identification. In the example given, the results were inconclusive. When two components contribute to the observed absorption one could theoretically resolve that if the absolute cross-sections are known. Figure 10 legend and trace colors should be consistent. I don't see a red trace.

P12, L33 "that peaks"

P13, L15: "observed chemical complexity" – in general measuring true complexity is likely beyond scope of any one study? The main benefit of this study is a wealth of new chemical information and tying that to absorption. For the former point, if not already done, it might be worth flagging which peaks are new. Since many commonly observed species were not seen, but some new ones were, perhaps a good topic for the conclusions is if this approach occupies a unique niche in "chemical space"?

P13, L31-32: This could be taken as: this study is the first to see non-lignin-derived entities in BBOA? Seems unlikely, clarify?

On SI:

Position/timing of cans is not clarified, etc.

---

## Referee Comment (RC2) · Anonymous Referee #2 · 6 Nov 2017

The Fleming et al. manuscript reports on chemical speciation of fine particulate matter (PM2.5) emitted from cookstoves. Two types of stoves were evaluated, as well as two types of fuel (dung and brushwood). The stoves were operated under realistic conditions (e.g., traditional meals, local cook). Samples were collected onto PTFE filters and were analyzed off-line using advanced high-resolution mass spectrometry techniques. In addition to expanding the list of reported compounds in biomass burning PM2.5 samples, brown carbon (BrC) chromophores were identified and mass absorption coefficients (MAC) were estimated. There are many strengths of this manuscript, including the effort to represent real world conditions, the application of advanced instrumentation, and the novelty of the reported results. This study likely represents the

most comprehensive analysis of the chemical composition of brushwood- and dung-generated primary PM2.5. The manuscript is well written and should be of interest to biomass burning, air quality and climate communities. It is thus appropriate for publication in ACP. Minor technical and editorial comments are provided below.

Technical:

Sample collection: have particle losses through the aluminum tubing been characterized? Would any size dependent losses bias the results?

MAC estimation: Can some uncertainty bounds be given for, 1. use of a separate filter for total mass and 2. range of estimated extraction efficiencies? Fig. 8 should include some uncertainty bounds/shading.

EF approximation: Is it reasonable to assume the peak abundances are proportional to mass concentrations? It would be useful to provide support for this assumption in either the manuscript or the supporting information. Given the uncertainties and required caveats, is there adequate justification for reporting emissions factors? Relative peak abundance may be more appropriate.

Nano-desi results (p. 7): The fractions of CxHyOzNw are relatively similar within and across fuel and stove types, with the exception of the brushwood sample RE007. That sample also appears to have a higher moisture content. Can any linkages between moisture content and PM2.5 chemical composition be made? Does this also influence the presence of BrC chromophores and can the differences between the values reported in this paper and in prior work be attributed in part to difference in fuel moisture (e.g., p. 11, line 17-20)?

Levoglucosan: The suggestion that levoglucosan may be a "good" tracer for the two fuel types may be misleading in the context given (i.e., present in less than half of the dung and brushwood/chulha samples). It is suggested to revise this statement.

Editorial:

[Figure]

The motivation for this work, as articulated in the introduction, is a bit unclear. There is quite a bit of discussion on the health implications of solid fuel use in cookstoves, and it is noted that the work was done as part of a larger study documenting the contribution of household combustion to ambient pollution (p. 4, line 4); however, the focus on MAC and BrC chromophores implies a greater relevance to climate. There is little to no discussion on the health implications of the identified compounds and no discussion of the local to regional implications of the findings (e.g., whether or not the MAC values and emissions factors are significant to suggest regional climatic influence).

p. 2, line 9-10: The clause "of pregnant women" after infants is a bit strange as written. Does this mean that exposure is through the mother? If so, one possible revision could be: "infants of women exposed while pregnant".

p. 2, lines 25-28: The discussion of estimated EFs from the Stockwell et al. manuscript is awkward as written. Revision is recommended.

p. 3, line 33: "prescribed" instead of "prescribing" ?

p. 5, line 50: "O"/oxygen does not need to be defined for DBE equation

p. 6, line 20: Remove "the" after "Since"

p. 13, line 3: SIC is undefined

Fig. 3: is confusing and provides little to no additional information beyond other figures and tables. Authors should consider removing it.

Fig. 5: "terpenes" is misspelled in figure legend

---

## Author Comment (AC1) · 27 Dec 2017

*Comments by reviewer #1 (Dr. Yokelson) are reproduced in the sans-serif font below. All comments have been numbered by us for convenience. Our responses follow each comment in a blue, italicized, serif font.* Text additions to the manuscript, for example, significantly modified sentences, appear in the manuscript in red color. *Deletions from the manuscript are not explicitly shown but are described in the responses below. Minor editorial edits to the text are not explicitly shown to prevent a cluttered view.*
* * *
Review of Molecular Composition of Particulate Matter Emissions from Dung and Brushwood Burning Household Cookstoves in Haryana, India, by Fleming et al.

By Bob Yokelson

The authors have successfully identified numerous individual chemical species that absorb UV light in an important, but under-studied type of biomass burning aerosol (wood and dung cooking fires). The authentic cooking fire samples are difficult to obtain and the author's samples were subsequently analyzed off-line with a unique, extremely powerful array of coupled optical and mass-spectral techniques. The team has a great deal of relevant expertise in all aspects of the study and it includes leading practitioners of these advanced analysis techniques. The large body of data represents a significant investment of effort and the work should definitely be published. With some straightforward improvements the paper could be very good. I provide an overview followed by specific suggestions referenced to page and line number in order of appearance.

Overview:

More information is needed about the sampling, calculations, and error budget in the paper. A few additional sampling details and equations are given in the SI, but more are needed and should be in the main paper. Currently, the emission factors are said to be "orders of magnitude" upper limits, but it's proposed via limited comparisons that the MAC are "OK" – though they are given without an error estimate. The final MAC seem potentially "high" to me and their application should be clarified. Along with more detail on the sampling, the associated uncertainty in each step should be discussed near the beginning of the paper so the context and limitations are clear as the discussion is read, rather than reader wondering and only potentially finding out later in the paper. I suggest a clear and commonly-used format would be to clarify how each step is done and discuss/estimate individually the different sources of error of each of these steps in order. For instance:

1. What were the inlet positions w.r.t. to the fires? Flaming produces a greater vertical velocity and flux of products than smoldering. Gas-particle partitioning is proportional to particle concentration and is also temperature dependent. Further from the source the smoke has diluted and cooled to some degree and vertical velocity doesn't need to be measured. A detailed diagram with distances, temperatures, concentrations (description of any dilution?), etc should be given in the paper.

*Figures S2.1 and S3.1 have been added to the supplementary information section, showing the dilution system and the positions of inlets with respect to the stoves, respectively. The probe position was fixed directly above the stove to allow for normal cooking activities, but still compare emissions. There are always tradeoffs in sampling emissions from solid fuel use in households. Emissions tests in laboratories using controlled hoods and dilution systems etc. have the benefit of more controlled sampling, but use water boiling tests that have systematically been shown not to reflect those during actual cooking, the subject of this paper. The discrepancies between field testing during actual cooking and water boiling tests are large (Johnson et al., 2008) and thus we elected to sample during actual cooking.*

*Cooking occurred in a real village kitchen (what we would consider to be the porch of the home), which made it challenging to sample well-mixed smoke. Although, the smoke diluted into the surrounding air before entering the dilution system, and subsequently filters were collected at ambient temperature. We acknowledge there are common artifacts with the filter sampling, for example, some of the smaller molecules we are observing in the mass spectra would be in the gas-phase in a diluted plume but they get trapped on the filter if there is a lot of organic material collected.*

2. What were the relative positions of the canister and filter inlets? Were they close or in wellmixed smoke?

*The gases and particles were sampled from the same inlet stream. We added Figures S2.1 and S3.1 as well as a description of Figure S2.1, P4, L4-11 of supplementary information section to clarify our sampling approach. Gases were collected in a Kynar bag over the course of the whole cooking event from which the canisters were filled.*

 3. What was the relative timing of the canister and filter sampling? Canisters tend to fill quickly at a non-constant rate while the filters were acquired over the whole fire.

*Thank you for pointing this out. We agree that the description of the gas sampling was not very clear. Canisters sampled the average (not instantaneous) gas-phase emissions over the cooking event, similar to the filter (particle) samples. Figures S2.1 and S3.1, P4 L4-11 of supplementary information were added to clarify this.*

4. When the results from two filters are coupled, what was the spatial and temporal overlap of the filter collection?

*Results from different filters were combined for samples done under as similar conditions as possible in terms of fuel type, stove, approximate moisture content, and meal cooked. Cookstove BBOA samples were collected over the course of a month. This is explained in P8, L29-31 of the manuscript. The filters used for chemical analysis always occupied the same space in the sampling train, shown in Figure S2.1. In terms of particle mass measurements, the gravimetric and chemical analysis filters were collected at the same time (same cooking event), however at different locations in the sampling train. This is now better described on P5, L5 in the main paper as well as in the supplementary information, P4, L15-19.*

5. The authors evidently measured extraction efficiency once at 50%, but did not specify 50% of what (PM2.5, OA, etc)? Nor is it stated if the extraction efficiency is the same for all chromophores. Later in

paper it's stated that the extraction efficiency was "lower sometimes." How much lower? Maybe give a range and consult some studies where extraction efficiency has been estimated by comparison to some familiar term e.g. OA measured by AMS or conventional OC analyses.

*We will address this question in comment #9. We have now constrained the MAC values by incorporating sources of error.*

6. Ionization efficiency is maybe stated near the end of the paper to be higher for polar compounds, but some polar compounds that are ubiquitous in BBOA like levoglucosan were not seen in some samples. Can this be explained?

*In electrospray ionization mass spectrometry (ESI-MS) different analytes have to compete for charge with each other leading to different ionization efficiencies for the same compound depending on what else is present in the mixture. This is known as the "matrix effect". Such matrix effects are likely responsible for inconsistent observation of some of the less ionizable compounds such as levoglucosan. This explanation was added to the manuscript, P8, L22-26.*

7. Given the realistic limitations of any technique, in the introduction or after the fleshed-out experimental/error section then maybe a brief summary of the strengths and weaknesses of various approaches to understanding aerosol optical properties is in order to provide context for readers. For instance, my understanding is that laser-based techniques would be well-suited for measuring the overall absorption of real aerosol (after drying) that contains BC, BrC, and other species at specific wavelengths, but non-power law features are an issue in fitting the crosssection at unmeasured wavelengths, they certainly cannot identify individual compounds, and they have some uncertainty in differentiating between BrC and coating effects (e.g. Pokhrel et al., 2017). Extractions (I think) eliminate BC and coating effects so that only BrC is probed. Following with off-line analysis by broadband UV absorption, retention times, and exact mass is very powerful for measuring the true BrC spectral shape and compound identification. But quantification and how the MAC relate to real-world aerosol is not clear from the paper now.

*We believe that adding a detailed description of the advantages and disadvantages of different techniques would make the paper too long. A recent comprehensive review by Laskin et al. (2015) has a long section explaining the instruments used to measure optical properties, as well as their limitations and advantages. Nevertheless, we added several sentences to explain the difference between the absorption coefficients reported by different methods. For example, we stated in the introduction section that "methods that do direct measurements on aerosol particles without dissolving them report $MAC_{aerosol}$, whereas measurements on extracted material report $MAC_{bulk}$." Here and in the experimental section we refer readers to Laskin et al., (2015) for more information about other methods.*

8. In light of above, provide some at least brief, rough guidance on how the overall optical properties of real BC-containing aerosol (coating effects and all) could be estimated from the extraction results that are presented. It seems possible using independently measured BC emissions, but I did not find that in the paper.

*We have added text on P12, L16-24 to the manuscript, estimating MAC for the real aerosol based off the approach of Stockwell et al. (2016). Uncertainties were also added in the text on P12, L25-33. For AAE, we clarified that this is an AAE for extracted OC only, and clarified that the cited Chen and Bond (2010) AAE value was also for the extracted OC. We also added a statement suggesting that whole aerosol (extractable and non-extractable components) will have lower AAE by comparing to Stockwell et al. (2016) in P13, L4-6.*

9. Make sure to specify what is being measured throughout (e.g. the MACs are absorption of what per mass of what?) and provide uncertainties.

*We now more explicitly state how we calculate MAC on P12, L8, and throughout the paper (please see #7). MAC values are still calculated based off the assumption that 50% of the total PM mass was extracted, however, they are better constrained. Uncertainties were added to Figure 8 and in the text that incorporate a 40% relative error for extraction efficiency, as well as flow rates (10% relative error).*

10. BBOA almost certainly contains 1000's of species. The mass spectra obtained depend on variable detection limits and loading and natural variability makes the number of samples important. Thus, throughout the paper, I would refer to "observed" complexity, with actual complexity beyond the capabilities of current instruments.

*We agree that the actual complexity is beyond the capabilities of our instrument. We are only probing the extractable, ionized constituents observed across multiple trials of the same cookfire type. Furthermore, the instrument is not capable of distinguishing structural isomers. We added word "observed" before complexity throughout the manuscript.*

11. Some new species are observed, but many species commonly observed by other techniques are missing some or all of the time. It would be helpful to clarify which off-line compound identification techniques access which types of chemical space well and, if possible, the relative mass contributions to total OA measured by various techniques. This could be a sentence or two in the paragraph recommended in point 7 above.

*The following paper shows that nano-DESI is particularly sensitive to nitrogen-containing compounds, compared to similar-sized compounds that do not contain nitrogen. This is conveyed on P10, L4-5, but we added this additional reference that backs up this point specifically for nano-DESI. We believe that it is beyond the scope of this paper to explain complimentary off-line techniques; this is something that should probably be done in a future review.*

*Laskin, J., Laskin, A., Roach, P. J., Slysz, G. W., Anderson, G. A., Nizkorodov, S. A., Bones, D. L. and Nguyen, L. Q.: High-Resolution Desorption Electrospray Ionization Mass Spectrometry for Chemical Characterization of Organic Aerosols, Anal. Chem., 82(5), 2048–2058, doi:10.1021/ac902801f, 2010.*

12. Tracers and markers are not the same thing and these terms are frequently misused. Tracers are emitted in a narrow, well-characterized ratio to the observable of interest and can be used to quantify impacts at receptor sites. Markers are useful but typically emitted in highly variable amounts by unique sources and useful qualitatively.

*Thank you for defining these terms. We replaced "tracers" with "markers" in the subheading for section 3.2. Additionally, the paragraph discussing LG (P8 L22-26) was edited with these definitions in mind. We also use levoglucosan to show this technique will preferentially ionize other constituents, and therefore, sugars, and broadly lignin-derived compounds, are underrepresented/absent from the reported inventory of the observed species.*

Comments on text in order of appearance:

P1, L13: "organic particles" should be "organic matter in particles" or "organic aerosol" to allow for the possibility of internally mixed particles.

*"organic particles" was changed to "organic aerosols," and similarly on P1 L22-23.*

P1, L20: "selected "extractable"" or similar seems appropriate before "compounds" or "numerous" instead of "selected"

*"numerous" was added before "compounds."*

P1, L21: The fact that many of these species are newly observed will make the reader curious if they are newly observed because they are relatively rare or because most of the organic matter in BBOA was previously un-speciated. If that question can be answered it would be great to do so in the paper somewhere. A related fine point is that if most of the observed compounds are newly seen despite the fact that numerous other compounds have already been seen in BBOA, then it is unlikely that this study probes the true molecular complexity. This is an empirical, observed molecular complexity impacted by detection limits and loading.

*It is possible that the newly identified species represent a small fraction of the BBOA particle mass, but they show up prominently in the mass spectra because they are readily ionized by nano-DESI. However, there have not been many studies of chemical composition of smoke produced by burning of dung, so it is also conceivable that these species have not been reported before because of the paucity of observations. Future studies should attempt to quantify relative contributions of different classes of compounds to the particle mass. We agree that we are not able to probe true molecular complexity with this technique. In the paper, we clarify in all instances that this is observed molecular complexity.*

P1, L22: "stove-specific combustion conditions" might be better than "stove" if the stoves impact emissions by impacting the mix combustion processes?

*We agree that both stove and fuel types affect the combustion conditions, but here we are just listing the variables we had to work with changing for each cookfire.*

P1, L23: "emission factor" and "observed molecular complexity"

*"Emissions factor" was changed to "emission factor," and "observed" added to "molecular complexity."*

P1, L25-7: These MACs seem ~2x too high compared to other studies if they are referenced to g of PM2.5, especially as a lower limit? And it should be clear what they are for; real aerosol with BC included or just the BrC? If BC is added back to represent real aerosols, how is that done?

*We partly addressed this question when answering comments #7 and #8. When editing the text, we made it clear in the text that the MAC reported is just for the extractable potion of the organic matter, and does not include contribution from BC. Furthermore, we made it clear that it is MAC of the bulk material from which particles are made, and not MAC of the aerosol.*

P2, L12: "depends to some extent" since amount of PM and individual susceptibility to various toxins matter a lot.

*The addition was made.*

P2, L20: lifetime of BrC also important.

*We agree. However, since the focus of this paper is primary emissions of BrC; we opted to leave the discussion of BrC aging processes out.*

P2, L24: probably don't need same reference twice?

*The second reference was deleted.*

P2, L25: "a photoacoustic spectrometer" should be "photoacoustic extinctiometers (PAX)" (throughout) and "895" should be "870", L26: "cook" > "cooking"

*These changes were made.*

P2, L26-29: The EFs for BrC quoted from Stockwell et al are actually their EF Babs at 405 nm (not the same thing as explained next). These include absorption by BC and BrC. Different EFabs, tentatively for just the BrC are also provided though.

Some relevant background on Stockwell et al 2016 from the corresponding author:

Lack and Langridge (2013, Table 1) recommended an MAC for "BrC," but actually meant an MAC referenced to the mass of "BrC-containing OA." They clarified that their MAC was an average value for BBOA but that the MAC can range a lot. The OA MAC was later found to depend on BC/OA by Saleh et al 2014. We used the concept of an EF for "BrC" based on the Lack and Langridge MAC in Stockwell et al, but have since abandoned that terminology as we think it is too easy to misinterpret. Meanwhile, in the Stockwell paper cited here we tried to present qualifying text as follows: "The BrC mass calculated this way is considered roughly equivalent to the total organic aerosol (OA) mass, which as a whole weakly absorbs UV light, and not the mass of the actual chromophores. The MAC of bulk OA varies substantially and the BrC mass we calculate with the single average MAC that we used is only qualitatively similar to bulk OA mass for "average" aerosol and even less similar to bulk OA for non-average aerosol (Saleh et al., 2014). The BrC mass estimated by PAX in this way was independently sampled and worth reporting,

but the filters and mAMS provide additional samples of the mass of organic aerosol emissions that have lower per-sample uncertainty for mass. Most importantly, the optical properties from the PAX (SSA, AAE, and absorption EFs calculated as detailed below) are not impacted by MAC variability or filter artifacts."

Thus, our EF BC and the values listed just above are the best to compare to in Stockwell et al. Thanks to the authors, we rechecked our cooking fire table, found one error that only impacts the SSAs, and have posted a corrigendum. (We discovered that the SSA labels in Table 4, but not Table S8 had been reversed: it should be SSA 870 and then SSA 405 below that in Tab 4.)

Then finally, the more robust estimates of speciated PM2.5 mass from the same study that we alluded to in Stockwell et al., 2016 can now be found in Jayarathne et al. (2017) and are useful for comparisons with this work as will be pointed out.

Jayarathne, T., Stockwell, C. E., Bhave, P. V., Praveen, P. S., Rathnayake, C. M., Islam, Md. R., Panday, A. K., Adhikari, S., Maharjan, R., Goetz, J. D., DeCarlo, P. F., Saikawa, E., Yokelson, R. J., and Stone, E. A.: Nepal Ambient Monitoring and Source Testing Experiment (NAMaSTE): Emissions of particulate matter from wood and dung cooking fires, garbage and crop residue burning, brick kilns, and other sources, Atmos. Chem. Phys. Discuss., https://doi.org/10.5194/acp-2017-510, in review, 2017.

*Thank you very much for these detailed explanations. Your discussion brings up an excellent point that we did not make clear enough in the original version of the manuscript. All molecules absorb radiation to a different extent. Classifying some of them as chromophoric and other as not chromophoric is arbitrary. Therefore, MAC should always be calculated by normalizing the measured absorption coefficient by the mass concentration of all molecules present in the material, not just the ones that are arbitrarily designated as chromophores. This is a common approach followed in papers that report MAC measurements for extractable organic material, and also the approach followed in this paper.*

*For unfortunate reasons, MAC is used in the literature to denote two different quantities. One is mass-normalized absorption cross section of aerosols ($MAC_{aerosol}$), which is absorption coefficient of air containing dispersed aerosol particles divided by their mass concentration. $MAC_{aerosol}$ is particle size dependent. The other one is mass-normalized absorption coefficient of the material from which aerosol particles are made ($MAC_{bulk}$), which does not depend on particle size. To help minimize confusion between the two, we renamed our MAC into $MAC_{bulk}$ throughout the paper.*

*We elected to compare our results to measurements by Stockwell et al. (2016) by approximately converting our emission factors for $MAC_{bulk}$ to emission factors for $MAC_{aerosol}$. The new paragraph on pages 12-13 is dedicated to such a comparison.*

*The cited EFs are now correct in paper. P3,L11-17 was amended to clarify these are absorption EFs of total OC. We will compare with Jayaranthne in our next paper that deals specifically with EFs for $PM_{2.5}$ and EFs of individual VOCs.*

P2, L31-32: Just an observation that the dung is expected to have lower MAC than wood consistent with lower BC/OA per Saleh et al 2014.

*Thank you for the suggestion. A reference to Saleh et al. (2014) was incorporated into explaining Pandey's result.*

P3, L13: Should "The" which could imply "all" be "Many"?

*This change was made.*

L17-19: This second half of the sentence starting with "while" is unclear in and also ceanothus is from US NW.

*The sentence was clarified to the following.*

*"Fuels utilized in the FLAME studies were selected to represent North American wild fires, and the publications focus on non-woody biomass fuels such as detritus and litter as well as ceanothus from the US Pacific Northwest."*

P3, L21: N accounts for a small mass fraction of BB PM2.5 so N-containing organics are likely a small fraction of the total BBOA?

*We added "detected" in front of species.*

P3, L25: eliminate "The most" since that will quickly be dated?

*The suggestion was implemented.*

P3, L32: "more" or "additional"?

*"more" was changed to "additional."*

P4, L4-5: I'm sure this is worthwhile data regardless if it is the first or last data, but it seems unlikely this would be the first detailed study of brushwood or dung smoke since these sources have been studied intensively for > twenty years. A recent example that contains examples of historical references is Jayarathne et al. (2017). Simoneit et al have been characterizing smoke for > 20 years.

*"For the first time" was removed, and the sentence was reworded to the following.*

*In this study, the chemical composition of cookstove smoke produced from actual cooking events is probed in detail. Here we compare particle-phase constituents in cookfire smoke produced from different stoves and fuels.*

P4, L23-30: The pictures are useful, but Fig 1A currently shows a concoction of unidentified tubing and a diagram is also needed. Questions arise with some also in the overview. How did the authors ensure representative sampling of well-mixed emissions for all devices? For instance, was the data corrected for

the different vertical velocity in the smoke column above the fire? This is important because during flaming, the flux of emissions can be much greater than during smoldering. What were the flow rates and residence times in inlets, was any dilution used, were the downstream filters side by side or in series, were the pumps downstream of the filters, how were filters stored during the 25-30 hours not at -80 C (in a cooler with dry or blue ice or at ambient T, what was ambient T if relevant?), were backgrounds or field blanks taken, error in gravimetric analysis, etc? #1-3

*See answers to #1-3. Filters were stored at ambient temperature in the field and on the airplane, which is now described on P5, L10-11.*

P4, L24: "BBOA" > "PM2.5"

*This change was made.*

P5, L1-9: The next step after filter collection and before mass spec is filter extraction, i.e. how was extraction done, what is extraction efficiency compared to total BBOA and is it the same for all the species detected, etc? As written it sounds as if the filter is inside a capillary. It may not be possible to estimate extraction efficiency if done by a droplet flowing over the filter surface? But the bottom line should be clear.

*Nano-DESI extracts the material before it flows into the HRMS. It was shown in Roach et al., 2010a, 2010b that Nano-DESI dissolves all material extractable in the electrospray solvents (ACN/H$_2$O). We agree that the description of nano-DESI was not very clear. We have expanded the explanation of Nano-DESI.*

P5, L6-7: Why positive mode and are there species only seen in negative ion mode?

*Samples were only run in the positive ion mode. Smith et al. (2009) using same technique for BBOA found 1.5-4 times fewer peaks in the negative ion mode, suggesting the BBOA constituents more readily ionize in positive ion mode. They found largely the same species in the negative ion mode as the positive ion mode, but additional were found in the positive ion mode.*

P5, L7 What are MRFA and Ultramark, what is the relevance of the calibration species, how were the cal results applied?

*MRFA is a Met-Arg-Phe-Ala acetate salt (523.65 amu), and Ultramark 1612 (700<amu<1900) is a mixture of fluorinated phosphazines. Along with caffeine (194.19 amu), these standards are used to calibrate the mass accuracy of the HRMS over a wide m/z range. These compounds are usually used without explanations in the mass spectrometry literature, but we translated some of this detail into the sentence.*

P5, L8-10: Filter deposits may not be uniform. What percent of peaks were seen in only one sample or had S:N < 3. This is useful context that helps relate reported complexity to observed complexity, which is itself a subset of actual complexity.

*This is a common, unresolved problem for direct infusion ESI-MS. The recorded ESI-MS spectra routinely contain spurious peaks appearing in the ionization process at random m/z values. One way to identify the actual chemical species in the sample is to run mass spectra for the same sample multiple times. The probability of having spurious peaks appearing at the same m/z value is very low, so the peaks appearing in all three mass spectra must be genuine. Further, low signal-to-noise and/or unusually high peak FWHM are useful indicators of spurious peaks, and help to filter these peaks out. In summary, we only keep the peaks in the final table when we are confident they are not coming from mass spectrometer noise.*

P5, L12: Does Kendrick analysis with CH2 and "H2" base units help with O-containing species?

*As long as the species have the same parent molecule (this can include oxygen and other atoms), and only differ in CH2 units, one can use Kendrick analysis to link species in the families.*

P5, L13: "mass-calibrated"

*The change was made.*

P5, L16: About what percent of signal or number of peaks was above m/z 350?

*Peaks above 350 m/z were minority species both in number and signal. This was highly variable, but on average, 9% of the number of peaks was above 350 m/z, while 6% of the total signal was above 350 m/z. These averages are now included in the manuscript.*

P5, L20: K is normally the most abundant alkali metal in BB-PM (not Na). S and Cl can be highin dung (Hosseini et al and numerous other papers). Brief explanation of why not included?

*While K is abundant, potassium-organic adducts were not observed in the mass spectra. This is due to the relative affinity of organic molecules for $Na^+$ and $K^+$, where the binding energies of organic molecules to $Na^+$ are much larger compared to $K^+$. We do observe inorganic adducts containing potassium.*

*The majority of the organic molecules contained carbon, hydrogen, oxygen, and nitrogen. Adding the possibility of S and Cl did not change the assignments. The small percentage of unassigned peaks could probably be assigned with S or other elements, but adding sulfur alone did not allow us to assign more peaks or change the assignments. This information was added to the manuscript on P5, L4-6.*

P5, L22: No "O" in the formula, but it is in the explanation?

*Oxygen was deleted from the explanation.*

P5, L27: Why is the extraction solvent mixture different here, i.e. not containing water? What fraction of mass is extracted and of what types of compounds (in summary form)?

*For measuring MAC values, our goal was to extract as much organic material as possible. Therefore, we chose three solvents covering a range of polarities; ACN, DCM, and hexanes. On the other hand, an ACN/water mixture was always used for electrospray (including nano-DESI) because less polar solvents*

*do not work well in ESI. The fraction of mass extracted is unknown, and it is the largest source of uncertainty in MAC calculations. We tried to estimate extraction efficiency in a separate experiment with a small number of filters from the same campaign (different filters than the ones analyzed), and got varying results of 30-60%. Therefore, we calculated MAC assuming 50±20% extraction efficiency. We expect to extract a range of compounds with this solvent mixture, from very polar organic molecules, such as nitrophenols, to polycyclic aromatic hydrocarbons.*

P6, L16: It should be specified that this is an MAC for the extracted BrC only. Are the units (cm) consistent with the reported MAC (m)? "Cmass" is the "solution mass concentration" of what and how measured?

*This change was made. $C_{mass}$ refers to the mass concentration of all the extractable organics in the solution.*

P6, L17-18: Another filter collected where and when? Here is an example why the spatial and temporal overlap of the filter collection is important to describe in experimental section. How are PM2.5 on the other filter and "Cmass" connected?

*See answer to #4. After accounting for flows, the PM mass on the gravimetric filter is assumed to be the same as the chemical analysis filter. We multiplied this by the extraction efficiency (0.5) to get $C_{mass}$. Two sentences were added to the manuscript to make this more clear (P7, L4-7).*

P6, L20: So is the target mass reference for the MAC PM2.5 mass then?

*Yes, it is. See previous comments for the changes we made to make this clear.*

P6, L20-21: How do they do know the mass extraction efficiency was < 50% sometimes and why would a lower mass extraction mean the MACs are a lower limit? What if the extraction got all the chromophores, but only half the total mass, then would the raw MAC be a factor of two high? It seems the impact on the MAC would depend on the relative extraction efficiency of the chromophores and other constituents.

*Please see responses to #5 and #9 above. We no longer say that the MAC values are lower limits, because we have been able to constrain them with uncertainties. We agree that the MAC values are highly dependent on the extraction solvents and the particle constituents.*

P6, L21: Again it seems the AAE are for the BrC component only? Can the authors add a sentence on how their AAE and MAC can be adjusted to represent real aerosol that also contains BC?

*Please see response to comment #8.*

P6, L27: Abundance is usually used to indicate how many there are of an item. Peak area or height is usually used to estimate the signal strength or amount of compound. Should "abundance" be replaced by "area" or something else?

*In mass spectrometry, the current convention is to say "peak abundance". Use of "peak area" and "peak intensity" is discouraged. Therefore, we retained "abundance" throughout.*

P6, L27-29: So if I understand this and the SI right, the EFPM(total) was estimated separately (in a sparsely described experiment) and then EFPM(total) was partitioned among the peaks observed on the MS according their relative intensity. If so this should be stated as a sentence in the main text. It's a concern that the real EF could be ten times or even several "orders" of magnitude lower. Should the authors reconsider even reporting EF directly as such? Maybe it's safer to label them as "upper limit EF in the tables and figures to spare potential future misinterpretation? Also, can the error be reduced or the error budget be tightened up? For instance, the authors may detect some species that have been better quantified from these sources in other studies? Can they use their ratios to any overlap compound with the extensive quantitative analyses reported in Jayarathne et al 2017 and many others? Ratios to levoglucosan or PAHs for instance may be helpful to constrain the "EFs" to a realistic range?

*We agree that the calculations done in our initial submission were too approximate. We made the suggested additions, including P4, L15-19 and Figure S2.1 in the supplementary information, and P5, L4-5 in the main paper to clarify the $EF_{PM}$ total calculations. We no longer report the upper limit for the emission factors; it was changed to relative abundance.*

P6, L28-30 and Figure 2 comments: It's true that authors show non-identical spectra from the sources even though the two dung spectra look pretty similar. But proof of distinct signatures requires enough samples to quantify the variability in each source. Right now it's not clear if the method uncertainty is larger or smaller than natural variability or what the observed variability is. The authors should try to characterize observed source to source variation with some metric (# or % of unique peaks) and this is done to some extent later in the paper, which is good. As noted above, the y-axis label should be "Upper limit EF" or "approx. relative abundance" Relevant to earlier comments, the spectra are too simple to represent all the components of BBOA although more details might be revealed with a log scale.

*The results and discussion section starts broadly with Figure 2 showing that brushwood/chulha has the most observed $C_xH_yO_z$ (blue), and dung/angithi has almost nitrogen-containing peaks (purple and red). Dung/chulha is somewhere in the middle with many nitrogen-containing peaks as well as $C_xH_yO_z$ peaks. We think it is overwhelming to show all mass spectra in the main paper. Instead, Table S1.1 contains the same info for all samples to show that these truly are representative spectra, and you can see these differences apart from natural variability. This is explained on P7, L18-23. The y-axis was changed to relative abundance on Figure 2.*

P7, L8: Dung has much higher N-content than wood and that makes excellent sense based on known plant and animal physiology. The Gautam reference has much higher N than normal for wood. See Stockwell et al 2016 for dung and Coggon et al Fig 3 for wood or many other sources referenced in these papers.

*The Gautam reference was retained, since the wood used in this study is different (shrub wood rather than tree wood). Nevertheless, nitrogen content for wood and dung fuels from Stockwell et al. (2016) and*

*Hatch et al. (2015) references were also added to the text. This would indeed help explain increased N content in the dung smoke.*

P7, L8-11: This sentence doesn't quite make sense. Typo? Punctuation?

*The punctuation was changed.*

P7, L12: Could provide a few key citations on lignin pyrolysis – there are many. It would be interesting if the authors could show that the cellulose preferentially ends up in the gas phase? But maybe the method has low sensitivity for sugars (from cellulose), which are usually abundant in BBOA (Christian et al., 2010; Jayarathne et al., 2017)?

*We agree that references should be added here; we cited Collard and Blin (2014) and Simoneit (1993). Unfortunately, we cannot show that cellulose decomposition products preferentially end up in gas phase due to the limitations of our technique.*

P7, L 14: Dung and embedded grasses are both high in Cl content (Stockwell et al., 2016 and references there-in, especially Lobert et al., 1999) so expect high Cl from cooking with dung and ag residues as in Stockwell et al. 2014-2015 ACP papers.

*A sentence and the references were added to P8, L4-5.*

P7, L15-16: Clarify if these large peaks were included when the EF was partitioned?

*We no longer are reporting emission factors for particle-phase chemical species because they were too approximate.*

P7, L21-26: K is well-known to be enhanced in biomass (Table 1 in Hosseini et al., 2013) and K has a very long history as a biomass burning "tracer" (e.g. Sullivan et al., 2014 and references there-in). It is well-known that K is primarily emitted by flaming while levoglucosan is primarily emitted by smoldering. If the stoves had different flaming/smoldering ratios that could explain variability in K production.

*This is a good point that dung/chulha had a higher combustion efficiency on average compared to dung/angithi. We added this in the discussion as well as the references below, specifically, P8, L3-4, 15-17.*

Hosseini, S., Urbanski, S., Dixit, P., Li, Q., Burling, I., Yokelson, R., Johnson, T., Shrivastava, M. K., Jung, H., Weise, D., Miller, W., and Cocker III, D.: Laboratory characterization of PM emissions from combustion of wildland biomass fuels, J. Geophys. Res., 118, 9914–9929, doi:10.1002/jgrd.50481, 2013.

Sullivan, A. P., May, A. A., Lee, T., McMeeking, G. R., Kreidenweis, S. M., Akagi, S. K., Yokelson, R. J., Urbanski, S. P., and Collett Jr., J. L.: Airborne characterization of smoke marker ratios from prescribed burning, Atmos. Chem. Phys., 14, 10535-10545, doi:10.5194/acp-14- 10535-2014, 2014.

P7, L27-30: There is a difference between a tracer and a marker and levoglucosan (LG, a cellulose pyrolysis product) is the latter. If LG was a good tracer then it would be in all the samples and at reproducible amounts. LG is in fact normally found to be a major, but variable component of BBOA (Sullivan et al 2014; Jayarathne et al 2017; Christian et al 2010). If LG is missing from many of the samples that suggests detection issues that could cause some of the BBOA species to escape the analysis procedures used.

*Thank you for defining these. See our response to #12 for explanation.*

P7, L31 - P8, L2: There were more than three spectra of each stove/fuel combo, but three were chosen how? Then peaks not appearing in all three spectra of a combo were discarded why? Then the remaining peaks were scaled and the spectra for each stove/fuel combo were averaged together. I think that is the right order, which may be jumbled in the text?

*Sample selection was explained in more detail (P8-9, L29-32, 1), and sentences were reordered correctly. Peaks not appearing in all three sample runs were discarded to ensure they are real peaks (see earlier explanation). We added "ensuring reproducibility" to the end of the statement to convey this.*

P8, L6-12: First compounds found from all three cooking types are listed. Then compounds only found from brushwood are discussed. That I can follow. Then the compounds that are found in all dung cooking. But not unique to dung cooking? It's not clear here what the difference between sections 3.5.1 and 3.5.2 is. I.e. what is the difference between "common to" and "detected in all"?

*We agree it was confusing as written. It was changed to the following. We next show compounds common to dung cookfire emissions (Section 3.5.1, Table 2). Lastly, we discuss BBOA compounds detected in either dung/chulha and dung/angithi cookfires (Section 3.5.2).*

P8, L11-12: If the stove material caused the emissions it potentially might not matter what the fuel is since all fires heat the stove.

*This is an excellent point. This is why we were always careful to specify both the fuel and the stove in the text because both contribute to the compounds emitted in the smoke.*

P8, L14-17: The true number of constituents for all the PM types is much greater than observed so maybe just say ~we saw more peaks from A than B.

*This was clarified.*

P8, L15-30: Can this overview of the results section be re-phrased or re-organized to make it easier to follow?

*We tried different approaches to organizing data and their discussion but elected to retain the current one because it appeared to be the most logical to the authors.*

P9, L4-5: "detected elemental" and the relative insensitivity for sugars, or anything else needs to be discussed earlier – maybe in the introduction or at latest the experimental section.

*We kept our wording "Biases the elemental make up" instead of "detected elemental composition," because we feel both are correct. We now discuss this disclaimer on P8, L22-26 when we discuss levoglucosan.*

P9, L6-8: I don't believe the %N in Gautam et al, it goes against all the other studies I've seen dating back to Susott et al., 1996, unless Gautam et al included the foliage with their wood.

*We also find the measurements of %N in Gautam et al. hard to explain. This sentence was deleted from the manuscript. Please see above comment on Gautam et al. regarding the difference of fuels in the studies you are referring to.*

P9, L17: All biomass is ~25% lignin, 25% hemicellulose, and ~50% cellulose polymers though the monomer units differ.

*A sentence was added to emphasize this point (P10, L17-18).*

P9, L18: delete "at"

*This was corrected.*

P9, L18-19: by "commonly detected" do they mean by their group or are there references to other groups? There are numerous studies that characterized BBOA.

*The sentence was modified to the following.*

*This suggests that perhaps 20% of the compounds listed in Table 1 might be reproducibly detected in BBOA samples using ESI-MS, regardless of biomass type.*

P9, L20-21: "found" to "observed" better?

*This was corrected.*

P9, L25: "coniferyl" alcohol and that comes only from conifers whereas the others are unique to hardwoods or grasses, which are probably more relevant in India.

*We appreciate you pointing out this typo. It is now fixed.*

P9, L30: In the experimental overview, the range of ionization efficiencies could be provided? #6

*Unfortunately, information on relative ionization efficiencies of different species is limited, and the efficiencies depend too strongly on the matrix effects. We are not in a position to provide this information.*

P9, L31: Jayarathne et al also discuss species unique to dung

*Absolutely, we no longer say in the manuscript that we are the first to speciate PM$_{2.5}$ from dung cookfires.*

P10, L1: important to qualify "the observed chemical …was far more complex" etc

*This change was made.*

P10, L2: By "reproducibly" it means "seen" in all "n" samples, but not in the same ratio to total PM2.5? That should be clear if so.

*We are not reporting the ratio to total PM$_{2.5}$. We explain the average abundance of the peak as Low, Medium, or High described on P9, L2-4.*

P10, L5-6: This is a little hard to follow as the authors use "found in all dung …" in the section header, then next mention "detected exclusively" in one type or another, and then "combine all…" Maybe change "Hereafter" to "However" would help with the transition?

*"Hereafter" was changed to "however."*

P10, L10 - P11, L6; Figures 5 -7: Jayarathne et al 2017 and references there-in quantified numerous PAHs for similar fuels, which may be helpful to compare to.

*This is great complimentary work, however, with electrospray we can only detect heterocyclic polycyclic aromatic compounds or substituted PAHs, so we cannot easily compare to the data. In response to this comment, we are currently measuring PAHs by a more conventional method and we plan to attempt a comparison for PAH emission factors in a follow up publication.*

P11, L8: At the outset useful to state what these MACs represent: absorption due to extractables only per PM2.5 mass? I think the mass reference is not extract mass or mass of chromophores. It's important in the larger context if they are MAC that don't include the BC component or any un-extracted chromophores.

*See explanation in #8-9.*

P11, L8-11: "browner" OA from wood cooking makes sense empirically given the higher EC or BC to OA ratio for these fires per Saleh et al., 2014, Jayarathne et al., 2017. The latter reference also reports higher PM emissions from dung cooking than wood cooking in agreement with cited previous work. Their measured and cited EFPM2.5 from South Asia for wood and dung cooking may be useful to compare to.

*A sentence was added to interpret the results using these references (P12, L12-14). In the next paper we will discuss our EF$_{PM2.5}$ in more detail, and will compare our results to Jayarathne et al. (2017) then. EF$_{PM2.5}$ is only brought up here to put our MAC values in context.*

P11, L12: The MAC is the absorption coefficient per mass of PM. In order to convey the absorption per unit fuel consumption, "coefficient by" should probably be replaced by "emission factor of"

*We now include both absorption of coefficient per mass of PM as well as the approximate absorption coefficient per unit fuel consumption for the whole aerosol using the approach described in Stockwell et al. (2016). See explanation for #8.*

P11, L 11-13: On the overall absorption per unit fuel consumption. This was evaluated by Stockwell et al., (2016) for particles containing BC and BrC and, with higher uncertainty, for just the BrC component, with the following results (in m2 /kg) (variability also shown in reference).

Wood Dung

EF Babs 405 10.6 5.85

EF Babs-405-BrC 8.40 5.43

EF Babs 870 1.04 .197

Wood-burning aerosol absorbed about 5 x more per kg burned at 870 because of the higher BC emissions, but just 2 x more at 405 for the overall particles. More BrC absorption per unit fuel consumption was observed on average for wood, but within variability the amounts overlapped. Can the authors use BC/EC EFs for these fire types to get their own estimates for absorption EFs for real fires?

*Thank you, we incorporated this. Please see our response for explanation #8.*

P11, L14-21: Throughout this section, for this work and other work, the mass reference (PM2.5, OA, etc?) should be rechecked and specified and uncertainties for the MACs should be provided. It makes sense that the brushwood MAC is larger than the dung MAC given the BC/OA dependence of MAC described by Saleh et al. (2014) as noted above.

*Please see our response to #9. A sentence was added showing our results are consistent with Saleh et al., 2014 (P12, L12-14).*

At a cursory glance, it seems like the MAC values at ~400 nm that the authors selected for comparisons are on average higher than values in Lack and Langridge, the extensive tables in Olson et al 2015, or Bluvshtein et al., (2017). It could be helpful if the authors could include these studies in their discussion and comment on any implications there may be.

Olson, M. R., Garcia, M. V., Robinson, M. A., Van Rooy, P., Dietenberger, M. A., Bergin, M., and Schauer, J. J.: Investigation of black and brown carbon multiple-wavelength-dependent light absorption from biomass and fossil fuel combustion source emissions, J. Geophys. Res.: Atmos., 120, 6682-6697, doi:10.1002/2014JD022970, 2015.

*In order to compare apples to apples we elected to only compare our measurements to studies that reported $MAC_{bulk}$. We cannot easily compare our results to studies reporting $MAC_{aerosol}$ (such as the paper by Olson mentioned here) because we would need to know the size distribution of particles, as well as their real refractive index for such a comparison.*

P11, L22-26: If my understanding is correct, one advantage of extraction techniques is that the BC is eliminated along with uncertainties in BrC attribution due to BC coatings or AAE. If that is right, the authors should clarify this is an AAE for the BrC only. Also, the AAEs can depend on the choice of wavelengths fit with a power law and this study has much better wavelength coverage than just the 2-3 wavelengths commonly used in optical in-situ approaches. The overall AAE with BC included is also important to describe real in-situ aerosol. The AAE with BC included may be lower since the AAE of BC is near 1. Stockwell et al obtained AAEs of 3 and 4.6 for wood and dung, respectively using an optical in-situ approach on aerosol containing BC. Can the authors estimate overall absorption values for real intact BC-containing aerosol from their extract values? It seems doable using EFBC and BC optical properties.

*We specified that this AAE is for OC/BrC only. We added P13, L4-6 to state that in situ AAE for cooking aerosol will be lower.*

P11, L25: Why is BC mentioned? Per above, isn't it eliminated in the extraction process? It's not clear why the authors say the following: "However, the observed absorption can be definitively attributed to BrC since AAE values of 2 or greater indicate that light absorption comes from BrC as opposed to BC (Kirchstetter et al., 2004; Laskin et al., 2015)." Pure uncoated BC has an AAE near 1, but any aerosol AAE can have some contribution from BC if BC is present. I think the definitive attribution comes from the fact that BC and BC coating effects are eliminated in the extraction.

*This sentence was deleted; we agree BC should not be mentioned.*

P11, L27: Move first sentence to P12, L17?

*We decided to keep it as is, because these paragraphs still focus on chromophore identification, even if we are not yet discussing the chromophores themselves.*

P12, L4: "lignin-derived"?

*This change was made.*

P12: L3-16: This is great stuff. I wonder if retention times were measured for standards to support identifications.

*At this time, standards were not used to support identifications (because we did not know what to expect when running the mass spectra). The reference absorption spectra had to be consulted during the data analysis stage.*

P12: L22-28: On line 24, why were the early eluters ignored? Combining line 22 and line 26, is the conclusion that polar compounds tend to be detected more efficiently, but elute earlier and may be ignored? Minor tweaks to text could likely clarify this section.

*We changed the wording of this sentence, and cited Lin et al., 2016, which explains this in more detail.*

P12, L29 – P13, L2: It's neat that absorption features can be used with mass and retention times to support compound identification. In the example given, the results were inconclusive. When two components contribute to the observed absorption one could theoretically resolve that if the absolute cross-sections are known. Figure 10 legend and trace colors should be consistent. I don't see a red trace.

*It is theoretically possible to calculate the relative contributions of each chromophore, in this case, ethyl-3-methoxybenzoate and veratraldehyde. If the focus of the study was to compare to standards, and get quantitative information about the contributions of different structures, we would definitely do this analysis. First, we do not have a full reference spectrum of veratraldehyde. Additionally, the solvents for the reference spectra are usually different from the mobile phase (mostly $H_2O$). As for the red trace, we believe this is an error that occurred in the proofs (it was red in the initial submission but changed color in the ACPD document). We will need to ask the editor about this, and we appreciate you pointing this out.*

P12, L33 "that peaks"

*The change was made.*

P13, L15: "observed chemical complexity" – in general measuring true complexity is likely beyond scope of any one study? The main benefit of this study is a wealth of new chemical information and tying that to absorption. For the former point, if not already done, it might be worth flagging which peaks are new. Since many commonly observed species were not seen, but some new ones were, perhaps a good topic for the conclusions is if this approach occupies a unique niche in "chemical space"?

*We agree that true chemical complexity cannot probed by this technique alone or this study alone. We changed wording to "observed chemical complexity".*

P13, L31-32: This could be taken as: this study is the first to see non-lignin-derived entities in BBOA? Seems unlikely, clarify?

*We agree and clarified the sentence.*

On SI:

Position/timing of cans is not clarified, etc.

*Please see the responses to questions #1-3.*

---

## Author Comment (AC2) · 27 Dec 2017

*Comments by reviewer #2 are reproduced in the sans-serif font below. Our responses follow each comment in a blue, italicized, serif font.* Text additions to the manuscript, for example, significantly modified sentences, appear in the manuscript in red color. *Deletions from the manuscript are not explicitly shown but are described in the responses below. Minor editorial edits to the text are not explicitly shown to prevent a cluttered view.*
* * *
The Fleming et al. manuscript reports on chemical speciation of fine particulate matter (PM2.5) emitted from cookstoves. Two types of stoves were evaluated, as well as two types of fuel (dung and brushwood). The stoves were operated under realistic conditions (e.g., traditional meals, local cook). Samples were collected onto PTFE filters and were analyzed off-line using advanced high-resolution mass spectrometry techniques. In addition to expanding the list of reported compounds in biomass burning PM2.5 samples, brown carbon (BrC) chromophores were identified and mass absorption coefficients (MAC) were estimated. There are many strengths of this manuscript, including the effort to represent real world conditions, the application of advanced instrumentation, and the novelty of the reported results. This study likely represents the most comprehensive analysis of the chemical composition of brushwood- and dung- generated primary PM2.5. The manuscript is well written and should be of interest to biomass burning, air quality and climate communities. It is thus appropriate for publication in ACP. Minor technical and editorial comments are provided below.

Technical:

Sample collection: have particle losses through the aluminum tubing been characterized? Would any size dependent losses bias the results?

*We have not characterized particle losses in aluminum tubing, but we expect it to be similar to copper or stainless steel tubing. The length of tubing was minimized in the set up to reduce particle losses. However, since small particles tend to diffuse to the walls, this could be an issue for $PM_{2.5}$.*

*There are practical limitations in sampling emissions from solid fuel use in households. Emissions tests in laboratories using controlled hoods and dilution systems etc. have the benefit of more controlled sampling, but the use of water boiling tests have systematically been shown to not reflect those during actual cooking, the subject of this paper. We anticipate that the discrepancies between field sampling during actual cooking and water boiling tests are much larger than one would expect from losses of small particles to the walls of the tubing. Thus, we chose to sample during actual cooking events with the associated constraints.*

MAC estimation: Can some uncertainty bounds be given for, 1. use of a separate filter for total mass and 2. range of estimated extraction efficiencies? Fig. 8 should include some uncertainty bounds/shading.

*Thank you for this suggestion. Uncertainties were added to Figure 8 and in the text that incorporate a 40% relative error for extraction efficiency, as well as flow rates (10% relative error).*

EF approximation: Is it reasonable to assume the peak abundances are proportional to mass concentrations? It would be useful to provide support for this assumption in either the manuscript or the supporting information. Given the uncertainties and required caveats, is there adequate justification for reporting emissions factors? Relative peak abundance may be more appropriate.

*We agree that the emission factors provided could be biased given different ionization and extraction efficiencies for different constituents. Therefore, we have changed the y-axis on Figure 2 to relative ion peak abundance (which is measured explicitly in the experiment).*

Nano-desi results (p. 7): The fractions of CxHyOzNw are relatively similar within and across fuel and stove types, with the exception of the brushwood sample RE007. That sample also appears to have a higher moisture content. Can any linkages between moisture content and PM2.5 chemical composition be made? Does this also influence the presence of BrC chromophores and can the differences between the values reported in this paper and in prior work be attributed in part to difference in fuel moisture (e.g., p. 11, line 17-20)?

*We were hoping to see this connection as well. However, in the samples we collected, binned into wet and dry fuels, there was not a clear trend with moisture content and $PM_{2.5}$ composition.*

Levoglucosan: The suggestion that levoglucosan may be a "good" tracer for the two fuel types may be misleading in the context given (i.e., present in less than half of the dung and brushwood/chulha samples). It is suggested to revise this statement.

*We agree this was confusing. Levoglucosan should have been seen in all samples, however, the chemical constituents compete for charge in direct infusion ESI, and therefore we do not see it in all samples. We have added this explanation on P8, L22-26. We have amended the concluding statement on P8, L26-27, where we say levoglucosan serves as a marker rather than a tracer.*

Editorial:

The motivation for this work, as articulated in the introduction, is a bit unclear. There is quite a bit of discussion on the health implications of solid fuel use in cookstoves, and it is noted that the work was done as part of a larger study documenting the contribution of household combustion to ambient pollution (p. 4, line 4); however, the focus on MAC and BrC chromophores implies a greater relevance to climate. There is little to no discussion on the health implications of the identified compounds and no discussion of the local to regional implications of the findings (e.g., whether or not the MAC values and emissions factors are significant to suggest regional climatic influence).

*The health effects of particulate matter as they relate to chemical constituents from combustion are largely unknown. For example, cigarette smoke is now known to have 1000s of compounds that have various levels of toxicity. We always look for the usual suspects, for example PAHs, but the particle-phase is much more complex. It is essential to characterize this complexity before we can even start correlating the chemical composition to health effects. We are not in a position to evaluate the health effects of the smoke, but we recommend to future researchers to correlate newly observed organics with health effects (P15, L16). On the contrary, we do have access to methods that allow us to characterize the optical*

*properties of cookstove particles, and so we do this in the manuscript. Since local to regional implications of the findings involve many other factors, including the effects of cloud formation, secondary organic aerosol formation, as well as chemical aging of particles. These effects are the subject of more detailed atmospheric modeling which is not covered in this paper, but is forthcoming.*

p. 2, line 9-10: The clause "of pregnant women" after infants is a bit strange as written. Does this mean that exposure is through the mother? If so, one possible revision could be: "infants of women exposed while pregnant".

*We took your suggestion on wording.*

p. 2, lines 25-28: The discussion of estimated EFs from the Stockwell et al. manuscript is awkward as written. Revision is recommended.

*The text was reworded for clarification purposes.*

p. 3, line 33: "prescribed" instead of "prescribing" ?

*The change was made.*

p. 5, line 50: "O"/oxygen does not need to be defined for DBE equation

*The change was made.*

p. 6, line 20: Remove "the" after "Since"

*The change was made.*

p. 13, line 3: SIC is undefined

*It is defined on P13, L24.*

Fig. 3: is confusing and provides little to no additional information beyond other figures and tables. Authors should consider removing it.

*Respectfully, we have elected to keep Figure 3. It may seem unnecessary to careful readers, however, it serves as a visual for the construction of the paper that readers can refer back to as they are reading the results and discussion.*

Fig. 5: "terpenes" is misspelled in figure legend
*Thank you for catching this. The change was made.*

---

## Author Response (AR2)

*Our response to the editor is in italics. Additions to the manuscript and supporting information are shown in green. Deletions are not shown.*

Dear Authors:
Thank you for your careful consideration and responses to the referee's comments. Following your attention to the minor edits below, I accept this for publication.

Page 3 line 14-15: I find this sentence difficult to read since the part about "just organic compounds" requires some assumptions regarding MAC(aerosol) that are explained afterwards. Revising this sentence would improve the readability of the manuscript.
*The sentence was amended to the following.*
*Stockwell et al., 2016 utilized photoacoustic extinctiometers (PAX) to conduct in situ absorption measurements at 405 and 870 nm, resulting in particle absorption coefficients from cook fire emissions in Nepal.*

Page 3 lines 23-24: I find these two sentences hard to interpret because the ratios are first given as BC/OA then switched to OC/EC. I suggest keeping the ratios consistent (i.e. switch the second to EC/OC). Additionally, since two references are cited, clarifying which study "In that particular study," is in reference to will increase readability.
*We changed OC/EC values to EC/OC. We also clarified that EC/OC values given are from Pandey et al., 2016.*

Page 8 line 20: I believe you are missing a reference to the angithi stove in the sentence "The Chulha stove produced meals for people or animal fodder, respectively."
*This sentence was corrected.*

Page 9 line 12/Figure 3: The section and table numbers need to be updated to be consistent in the text and the figure. For instance, the table containing the Brushwood/chulha peaks is Table S4.1. There may be other inconsistencies as well.
*This appears to be correct. Figure 3 was updated, and all figure and table numbers were rechecked.*

I suggest considering combining sections 3.5.1 and 3.5.2 into just section 3.5. The heading titles used for 3.5.1 and 3.5.2 are confusing for me and 3.5.1 is very short.
*The suggestion was taken. The text was amended as well as Figure 3.*

Figures 4 & 9: The lines outlining some of the subpanels are distracting. Please remove.
*We believe we have addressed this.*

Figure 5: The colors of lines are hard to distinguish
*The color of the polyenes trace was changed to green for more contrast.*

Supplement:
Page 4 line 3: Should this reference Figure S2.1 not S3.1?
*Thank you for catching this, the change was made.*

Section S4: Please revise the section heading since Table S4.1 includes brushwood.
*We made this addition.*

[revised manuscript text omitted]

PM$_{2.5}$ emission factors are briefly mentioned when comparing absorbance by particles from different cookfire types (Section 3.6). Here we explain how they were calculated. Figure S2.1 shows the sampling lines used to collect emissions in this study. Emissions flowed through a PM$_{2.5}$ cyclone and subsequent quartz filter to remove particles, so that gases were
5   collected over the entire cooking event in an 80 L Kynar bag (Gases sampling line, Figure S2.1). After pumps were turned off, a whole air sample (WAS) of average gas-phase emissions over the cooking event was collected from the Kynar bag. Stainless steel canisters (2 L), evacuated and prepped prior to the trip, were used to collect WAS. The background WAS sample was collected as a grab sample in the kitchen before cooking began for the day. One background sample was collected per day and that measurement was used for all experiments that day. Ideally background samples should be an
10   integrated sample collected at the same time as the sample. However, we were limited in the number of cans brought to India.

**Figure S2.1. Diagram of sampling lines used in the study.**

[Figure]

A separate filter reserved for gravimetric analysis was used for fine particle emissions measurements (Teflon A).
15   These filters were pre-weighed on a Cahn-28 electrobalance after equilibrating for a minimum of 24 hours in a humidity and temperature-controlled environment (average temperature 18.9 degrees Celsius, standard deviation 0.4 degrees Celsius, average relative humidity 64%, standard deviation 7%). This PTFE filter collected cookstove emissions on a separate line than the filter analyzed by nano-DESI-HRMS and HPLC-PDA-HRMS techniques (Teflon B). Another gravimetric filter was collected in the background during the cooking event and was equilibrated and weighed in the same way. The masses for the
20   background and sample filters were utilized after accounting for the difference in flow rates. Then the background mass was subtracted from the sample mass to obtain the mass of PM (m$_{PM}$) in equation (1) below.

$$\frac{EF_{PM}}{EF_{CO}} = \frac{m_{PM}/V_{air}}{m_{CO}/V_{air}} \tag{1}$$

The concentration of CO was measured using WAS samples. The WAS samples were taken back to UCI where they were injected into a GC-FID with a Ni catalyst that converts CO into detectable $CH_4$. Other gases were also detected using a GC system comprised of 3 gas chromatographs equipped with 5 columns (DB-1, Restek 1701, DB-5ms) and detectors (FID,

5    ECD, MS). A complete list of gaseous emission factors will be reported in a separate manuscript.

$EF_{CO}$ was produced using the carbon-balance method. This method traces carbon in the form of emitted $CO_2$, CO, $CH_4$, other hydrocarbons, and PM and utilizes the relative concentrations of these compounds to evaluate emission factors. The total gas-phase carbon emissions were approximated with the concentrations of 86 gases measured using WAS. The ratio of the mass concentration of carbon in CO ($C_{CO}$) to the total mass concentration of detected gas-phase carbon was calculated

10    using equation (2).

$$C_{CO}\ emitted\ (g) = \frac{C_{CO(g\ m^{-3})}}{\sum_1^{86} C_1 + C_2 + C_3 + \cdots + C_{86}} \cdot C_T\ (kg) \cdot \frac{1000\ g}{1\ kg} \tag{2}$$

In equation (2), $C_i$ represents the mass of carbon in compound i per $m^3$ of air. $C_T$ specifically refers to the net mass of carbon

15    in the fuel and is adjusted for ash and char carbon. The carbon content of the fuel was taken to be 33% for buffalo dung and 45% for brushwood fuels based on standard values from Smith et al., 2000. Carbon in ash was estimated as 2.9% and 80.9% of the measured char mass for dry dung and dry brushwood, respectively (Smith et al., 2000). Then, we calculated $EF_{CO}$ using equation (3),

$$EF_{CO}(^{g\ CO}/_{kg\ fuel}) = \frac{C_{CO}\ emitted(g) \cdot \frac{28.01\ g}{12.00\ g}}{mass_{fuel}(kg)} \tag{3}$$

20    where $mass_{fuel}$ is the net dry fuel in kg burned for the cooking event.

**S3. Gas and PM₂.₅ collection details**

**Figure S3.1. Stoves used in the study, the *angithi* and *chulha*, are pictured. Stove measurements and distances from the stoves to the inlet probes are found in the tables below.**

[Figure]

| Chulha measurements | Distance (cm) | Angithi measurements | Distance (cm) |
|---|---|---|---|
| Inner height | 25 | Inner diameter | 44 |
| Inner width | 21 | Outer diameter | 50 |
| Inner depth | 23 | Outer height | 20 |
| Width of walls | 4.6 | Height from inside bottom of stove to probe inlets | 83 |
| Top of *chulha* to probe inlets | 58 | Top of *angithi* to probe inlets | 64 |

**S4. Species exclusively detected in brushwood/*chulha*, dung/*chulha*, and dung/*angithi* cookfires**

[revised manuscript text omitted]